# CRITICAL CONFABULATION:
# CAN LLMS HALLUCINATE FOR SOCIAL GOOD?

**Peiqi Sui**
McGill University

**Eamon Duede**
Purdue University

**Hoyt Long**
University of Chicago

**Richard Jean So**
Duke University

## ABSTRACT

LLMs hallucinate, yet some confabulations can have social affordances if carefully bounded. We propose **critical confabulation** (inspired by *critical fabulation* from literary and social theory), the use of LLM hallucinations to "fill-in-the-gap" for omissions in archives due to social and political inequality, and reconstruct divergent yet evidence-bound narratives for history's "hidden figures". We simulate these gaps with an open-ended narrative cloze task: asking LLMs to generate a masked event in a character-centric timeline sourced from a novel corpus of unpublished texts. We evaluate audited (for data contamination), fully-open models (the OLMO-2 family) and unaudited open-weight and proprietary baselines under a range of prompts designed to elicit controlled and useful hallucinations. Our findings validate LLMs' foundational narrative understanding capabilities to perform critical confabulation, and show how controlled and well-specified hallucinations can support LLM applications for knowledge production without collapsing speculation into a lack of historical accuracy and fidelity.

"As an emblematic figure of the enslaved woman in the Atlantic world, Venus makes plain the convergence of terror and pleasure in the libidinal economy of slavery... [critical fabulation] attempts to redress it by describing as fully as possible the conditions that determine the appearance of Venus and that dictate her silence."

— Saidiya Hartman, (2008)

## 1 BACKGROUND

Large language models (LLMs) are prone to hallucinate, generating "plausible yet nonfactual" outputs (Huang et al., 2025). While typically treated as a failure mode, recent studies show that some hallucinations could in fact be valuable (Jiang et al., 2024; Hu et al., 2024; Taveekitworachai et al., 2024), especially a subset termed confabulations: a narrative-driven tendency to "fill in" missing information with self-consistent stories that bear close verisimilitude with reality (Sui et al., 2024). Confabulated texts are more narrative-rich and better match the patterns of human storytelling, a vital communicative and cognitive resource for sense-making (Herman, 2013). Thus, a principled

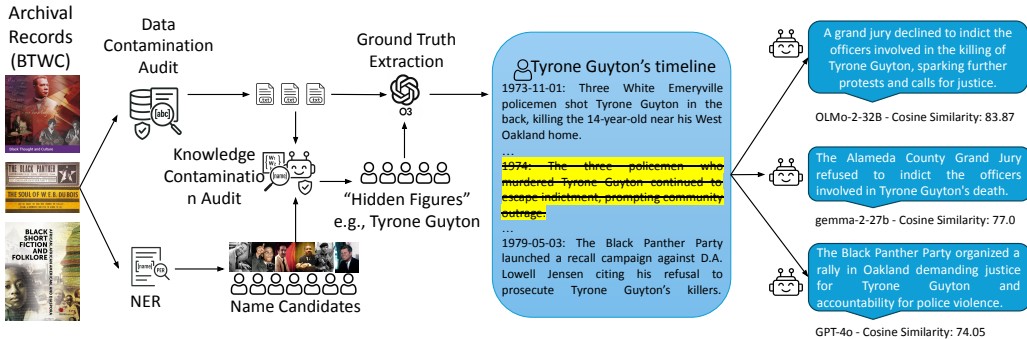

Figure 1: Critical confabulation as an open-ended narrative-cloze task for LLMs.

trade-off between strict factuality and alignment with the human behavior of storytelling may better support application settings that require interacting with LLMs as cultural agents instead of merely using them as tools.

If grounded in cultural or historical contexts, can a carefully controlled form of confabulation be harnessed for social good and useful knowledge production? Prior research shows this approach is feasible for AI applications across a range of fields, including computational creativity (Jiang et al., 2024), narrative exposure therapy (Lely et al., 2019; Li et al., 2023a), digital storytelling for cultural heritage (Shim et al., 2024), co-creative museum exhibitions (Fu et al., 2024), and bias mitigation (Kim et al., 2024). Importantly, these useful confabulations have clearly defined bounds that anchor them in existing evidence and differentiate them from unconstrained and underspecified hallucinations where the model must first speculate where the missing information is.

In the academic field of African American studies, Hartman (2008) introduces *critical fabulation*, the practice of using speculative storytelling to rectify omissions in historical archives due to social and political inequality, which has since become a field-defining methodology adopted in a wide range of humanistic historical case studies (Fuentes, 2016; Hartman, 2019; Hannah-Jones et al., 2021; Morgan, 2021). Many lacunae in historical records are the result of institutionalized violence and state and social repression that, like the Atlantic slave trade and its afterlives, has erased marginalized stories by structuring the standards for what is preserved or forgotten (Hartman, 1997; 2007). The harms of these lacunae are well-documented across the humanities and social sciences, described in postcolonial studies as the "epistemic violence" of identity and knowledge erasure (Spivak, 1988), in philosophy as the "hermeneutic injustice" of depriving the stories that allow one to make sense of their own lives (Fricker, 2007), and in anthropology as "archival silence" (Trouillot, 1995) and the "denial of coevalness" (Fabian, 2002).

Hartman argues that an empiricist and fact-based approach to historiography is problematic for addressing the long tail of archives—the "hidden figures" who were never afforded the privilege of historical documentation in the first place. Rather, treating record availability as a proxy for truth effectively overfits to the fraught criteria of what survives in the archive, a confounder that would further silence the archive's hidden figures and their divergent stories: gaps in the archive emerge when the standard of factual "rigor" privileges evidence-rich authorized accounts of history over necessarily sparsely documented stories that would be dismissed as insufficient.

As a form of socially reparative practice, critical fabulation aims to recover and reconstruct coherent stories from fragmentary records and humanize the hidden figures they omit, using speculative storytelling to read against-the-grain of the archives that have systemically excluded them. This often involves adopting marginalized points-of-view that contest and unsettle the authorized accounts of history, to better understand their limitations and interrogate "what might have happened". Crucially, the practice neither collapses speculation into facts nor licenses the invention of new historical realities; instead, it clearly defines a system of narrative ethics that bounds the scope of critical fabulation to high-fidelity counter-narratives grounded in the lacuna's context.

Drawing inspiration from how Hartman's framework augments low-resource settings with evidence-grounded human storytelling, we propose to leverage LLMs' extant behavior of confabulation to operationalize critical fabulation as **critical confabulation**: an AI storytelling workflow (Figure 1) for scaling the recovery of divergent yet evidence-bound narratives with well-documented affordances for useful knowledge production. Although critical fabulation is widely regarded as effective for enhancing the completeness of historical knowledge, its application has been limited by the vastness of archival corpora—faithful reconstruction of lacuna is labor-intensive because accountability to extant historical evidence requires meticulous parsing of dense and granular records. This tension motivates LLM-enabled approaches to critical fabulation that can 1) identify potential lacunae in large-scale archives, and 2) simulate multiple, evidence-constrained possibilities (rather than assert a single monolithic truth) to aid human scholars in the augmentation of historical knowledge. LLM-based methods build on prior work in digital humanities that demonstrates the compatibility between critical fabulation and computational techniques (Klein, 2013; Johnson, 2018; Risam, 2019; Risam & Josephs, 2021).

In this paper, we operationalize critical confabulation as a narrative cloze task (Chambers & Jurafsky, 2008), and use it as a proxy for fragmentary records to assess a wide range of LLMs. To build a realistic setting for evaluating the recovery of erased histories, we leverage the "Black Writing and

|  | BTWC | SEEN | UNSEEN | "Folktale" & "Story" |  |  | Count |
|---|---|---|---|---|---|---|---|
| Document count | 20686 | 4499 | 16187 | 4753 |  | Hidden figures | 156 |
| Total length (tokens) | 61.1m | 30.2m | 30.8m | 11.5m |  | Events (median) | 6 [4–9] |

|  |  |
|---|---|
| (a) | (b) |

Table 1: Dataset Statistics: (a) The BTWC corpus $\mathcal{B}$, (b) character-level ground truth timelines $\mathcal{T}(n)$

Thought Collection" (BWTC), a digital archive of mostly unpublished, primary sources related to black intellectual history and storytelling, as ground truth for a "lost" history unknown to the models. Our work advances the following research questions:

**R1: Can LLMs confabulate plausible candidates for archival omissions?** We find that while critical confabulation remains a very challenging task for state-of-the-art LLMs, most models can propose some plausible, evidence-bounded candidates for masked narratives. This capability positions them as viable co-creators to support humans in the practice of critical fabulation, and useful research tools for advancing digital humanities scholarship on archival silence's call for "technology of recovery" (Gallon, 2016) that aims to "animat[e] the mysteries of the past" (Klein, 2013). The effectiveness of historical confabulations could also broaden the cultural scope and social affordances of AI-assisted text restoration beyond the current domain-specific applications like ancient epigraphic reconstruction (Assael et al., 2022; 2025), diacritization (Gershuni & Pinter, 2022; Kharsa et al., 2024), and traffic surveillance (Gong et al., 2024; Pan et al., 2024).

**R2: What conditions shape, constrain, and enhance these confabulations?** We find that the effectiveness of event reconstruction is highly prompt-sensitive and improves with light supervision of specifying event type. Performance further varies on the event's structural attributes (length, position in timeline, timeline length), and most models perform best on events limited to biographical information and worst on events concerning intellectual history.

**R3: What makes an LLM good at critical confabulation?** Prompts that explicitly steer models towards hallucination reliably improve confabulation performance. GPT-5-CHAT is the overall leader, the only model to exceed 50% on most prompts and peaking at 59.7%. While larger variants usually outperform smaller ones within a family, smaller models can punch above their weight on this task: OLMO-2-7B attains the second-highest score under one prompt setting, and QWEN3-4B achieves among the strongest overall results.

## 2 TASK SETUP

**Problem view.** We divide the full process of critical confabulation (Figure 4) into **known unknowns** (lacuna reconstruction) and the more challenging **unknown unknowns** (lacuna detection). In this work, we cast critical confabulation as an open–ended narrative cloze task to first target the **known unknowns** of historical archives.

**Procedure.** For a hidden figure $n$ with a timeline (sourced from pertinent archival records $\mathcal{B}(n)$)

$$\mathcal{T}(n) \; = \; \langle (t_1, e_1), \ldots, (t_{m(n)}, e_{m(n)}) \rangle,$$

each element consists of a timestamp $t_i$ and a one–sentence event $e_i$. We simulate a lacuna in historical knowledge by replacing one event $e_m$ with the literal token [MASK]:

$$\mathcal{C}(n, m) \; = \; \mathrm{mask}\big(\mathcal{T}(n), m\big) = \langle (t_1, e_1), \ldots, (t_m, \texttt{[MASK]}), \ldots, (t_{m(n)}, e_{m(n)}) \rangle.$$

Given the timeline fragments $\mathcal{C}(n, m)$ and a fixed instruction prompt, a model $f_\theta$ must reconstruct the masked event $e_m$. Let $\mathrm{sim}_{\mathrm{emb}}(\hat{e}_m, e_m)$ denote the embedding–based semantic similarity between $\hat{e}_m$ and $e_m$. A reconstruction is counted as "correct" (sufficiently similar) if $\mathrm{sim}_{\mathrm{emb}}(\hat{e}_m, e_m) \geq \epsilon$, where $\epsilon$ is a model–specific, tunable threshold.

## 3 DATASET

### 3.1 DATASET STATISTICS

Our dataset for establishing a ground truth of hidden history is the "Black Writing and Thought Collection" (BWTC). This collection was curated by the ARTFL Project at the University of Chicago in collaboration with Alexander Street Press and is used with the permission of ARTFL. It is comprised of 20,686 documents dating from the early 1700s to modern times, and includes corpora of dramatic writing, fiction and folktales, and non-fiction works such as interviews, journal articles, speeches, essays, pamphlets, and letters. Although much of the material in the collection was previously published in print and includes writings by prominent black writers and intellectuals, a sizable portion is made up of previously inaccessible material that has not been widely distributed, including interviews and trial transcripts. We choose BWTC partly for this reason, as it ensures that a meaningful percentage of the collection is not included in the various book datasets used to train foundation models. Meanwhile, it is also well established that historical documents related to black history and culture have generally been under-represented in digital collections (Marcus & Carlson, 2018; Flinn, 2007; Prescod, 2017), making BWTC an ideal dataset to evaluate LLMs' foundational narrative capabilities for critical confabulation.

### 3.2 HAVE LLMS SEEN OUR UNPUBLISHED DATA?

Data contamination is a problem for LLM evaluations (Golchin & Surdeanu, 2024; Deng et al., 2024). In our case, contamination is especially problematic since the presence of our unpublished dataset in pretraining or instruction-tuning corpora would violate the assumption that LLMs have no prior access to this history, effectively collapsing the task of evidence-bounded confabulation into memorization. We focus our main experiments on fully open models with publicly released training data (e.g., the OLMO family), due to the absence of reliable methods for model-agnostic pretraining data detection. While recent work in this area reports optimistic results for popular approaches like membership-inference attacks (MIAs) (Shi et al., 2024; Zhang et al., 2024), subsequent studies challenge their robustness, finding that out-of-distribution MIAs often perform no better than random guessing (Maini et al., 2024; Duan et al., 2024).

Accordingly, we restrict our analysis to OLMO-2, the state-of-the-art fully open model whose public training data enable us to directly cross-search for our base corpora. To thoroughly vet OLMO-2 for our following experiments, we conduct a two-stage data audit on the sentence-level:

**String Search.**[1] To identify potential pretraining contamination in OLMO-2, we first perform an exact, sentence-level substring search that scans *every* sentence from the BWTC corpus $\mathcal{B}$ against the *entire* publicly released OLMO-2 training set $\mathcal{O}$ (Table 1). For each document $d \in \mathcal{B}$, we segment $d$ into sentences $S(d) = \{s_1, \ldots, s_n\}$. We then implement a classic Boyer–Moore substring matcher (Algorithm 1) for each query sentence $s \in S(d)$ against each candidate document $x$ in $\mathcal{O}$. The Boyer-Moore algorithm is selected for its efficiency and sublinear average-case runtime (Boyer & Moore, 1977), which is crucial at the scale of $\mathcal{B}$ and $\mathcal{O}$. For a document $d$, we compute the raw match count

$$\text{matches}(d) = \sum_{x \in \mathcal{O}} \sum_{s \in S(d)} \text{BM}(x, s),$$

where $\text{BM}(x, s) \in \{0, 1\}$ indicate whether $s$ appears contiguously in $x$. We designate $d$ with $>= 100$ matches as SEEN by OLMO-2. Overall, 21% of all documents in $\mathcal{B}$ are flagged as SEEN.

**Cosine Similarity Distributions.** As a sanity-check for Algorithm 1's SEEN and UNSEEN labels, we take inspiration from Oren et al. (2024)'s approach to detect contamination via model behavioral comparisons on input permutations, and perform an analogous probe on OLMO-2 in a next-sentence completion setting. Intuitively, we expect a memorization-induced performance gap – OLMO-2's continuations of sequences from $\mathcal{B}_{\text{seen}}$ should be more similar to their hold-out gold text than those from $\mathcal{B}_{\text{unseen}}$ to theirs such that $\forall i \in \mathcal{W} : \text{mean}_{d \in \mathcal{B}_{\text{seen}}}[\text{sim}_i(d)] > \text{mean}_{d \in \mathcal{B}_{\text{unseen}}}[\text{sim}_i(d)]$, where $\mathcal{W} = \{1, \ldots, K\}$ indicates a continuation window of $K$ sentences, and $\text{sim}_i(d) \in [0, 1]$ denotes a fixed scalar summary of the lexical similarity (e.g., a position-averaged similarity over $\mathcal{W}$).

---

[1]We perform our own search instead of using popular tools like WIMBD (Elazar et al., 2024), because it does not include the indices for OLMO-2's training data.

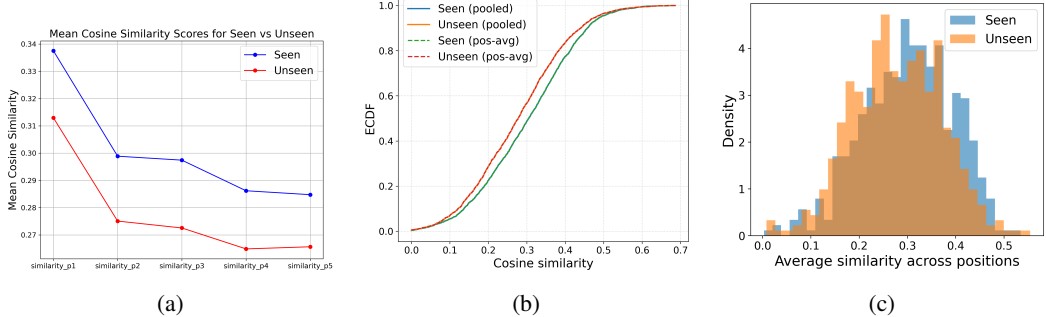

Figure 2: **Behavioral probe for contamination.** Visualizing the cosine similarity decay through (a) position-wise mean, (b) aggregate ECDF, (c) distribution across positions.

To validate this hypothesis, we construct a 1,000-document subset (500 randomly sampled from $\mathcal{B}_{\text{seen}}$, 500 from $\mathcal{B}_{\text{unseen}}$). For each document, we take the first 20 sentences as context and define five non-overlapping gold continuation windows of five sentences each, $p_1, \ldots, p_5$. We then generate continuations with OLMO-2, first given the 20-sentence context and subsequently with the addition of its own prior continuations, producing $\hat{p}_1, \ldots, \hat{p}_5$. For $\text{sim}(d)$, we compute a position-wise cosine similarity by fitting a *pairwise* TF–IDF vectorizer on each pair $(p_i, \hat{p}_i)$, and aggregate the scores by positions $i \in \{1, \ldots, 5\}$. Formally, $\forall i \in \{1, \ldots, 5\} : \text{mean}_{d \in \mathcal{B}_{\text{seen}}} \big[\text{sim}(p_i, \hat{p}_i)\big] > \text{mean}_{d \in \mathcal{B}_{\text{unseen}}} \big[\text{sim}(p_i, \hat{p}_i)\big], \hat{p}_i \sim \pi(P_\theta(\cdot \mid \hat{p}_{<i}(d)))$ where $\pi$ denotes greedy decoding for reproducibility, and $\theta$ are the parameters of OLMO-2.

The results support the hypothesis: overall, SEEN continuations exhibit consistently higher cosine similarity to the ground truth than UNSEEN (aggregate mean similarity is $0.3009$ vs. $0.2782$) (Fig. 2). This is consistent with a *memorization-induced advantage*: when the audit labels indicate prior exposure, OLMO-2's continuations hew closer to the held-out text. The monotone decay of the gap from p1→p5 accords with error accumulation as generations proceed, suggesting any advantage from memorization is strongest immediately after the observed context. While the absolute effects on similarity are modest, the signal is directionally robust across a diverse set of statistical tests (parametric, nonparametric, permutation) and survives multiplicity control.

Following the evidence that supports the validity of our SEEN/UNSEEN labels, we adopt a conservative document filtering policy: we remove all documents marked as SEEN from the dataset and conduct all subsequent analyses exclusively on $\mathcal{B}_{\text{unseen}}$. This filtering eliminates 21% of documents and likely discards some false positive documents from the string search, but it ensures that the remaining data functions as a reliable ground-truth proxy for *unseen* history and prevents spurious gains from memorization. Additionally, we use the dataset's metadata to filter out documents that are not in English; we also exclude documents under the genre "Folktale" or "Story", since their fictional nature makes them impertinent to our focus on historical confabulation.

## 4 GROUND TRUTH CONSTRUCTION

We leverage long–tail name candidates from $\mathcal{B}_{\text{unseen}}$ that are rare or absent in $\mathcal{O}$ as proxies for Hartman's "hidden figures," and, for each figure, we construct event-centric timelines from their pertinent documents as the ground truth for the narrative cloze evaluation. In other words, we first extract the **unknown knowns** from extant archives as a basis for the subsequent evaluation of the **known unknowns**.

### 4.1 WHO ARE THE HIDDEN FIGURES OF OUR DATASET?

Even after removing documents flagged as SEEN by our sentence-level audit, a model's parametric knowledge could still serve as priors for specific entities and names that do not appear in $\mathcal{B}_{\text{seen}}$ (Nasr et al., 2025). We therefore conduct a more robust search for "hidden figures" that are both in $\mathcal{B}_{\text{unseen}}$ and mostly absent from $\mathcal{O}$, using Aho & Corasick (1975).

**Name Candidates.** We curate a list $N^\star$ by extracting up to the top $10,000$ unique PERSON names from $\mathcal{B}_{\text{unseen}}$ (NLTK), then targeting the long tail by only keeping names within a lower frequency $< 51$. For downstream timeline quality, we also require that a name appear in at least three documents.

**Aho–Corasick Search.** We build a multi-pattern Aho–Corasick automaton $\mathcal{M}(N^\star)$ (Algorithm 2) and stream each document $x \in \mathcal{O}$ through $\mathcal{M}$, counting substring matches for both single- and multi-token names. For each $n \in N^\star$, we accumulate a global count $c(n)$ across $\mathcal{O}$. This avoids per-name passes and yields linear-time scanning in corpus size. To curb substring artifacts (e.g., "Ann" in "Annex"), we cache short surrounding snippets for low-frequency hits and manually spot-check a sample of those contexts. Using the same conservative threshold as our string-search audit, we declare SEEN-IN-$\mathcal{O}$ if $c(n) \geq 100$; otherwise we treat $n$ as an UNSEEN (n = 322).

**Manual Filtering.** To further clean the name list, we conduct manual filtering to remove names that only have trivial appearances in their documents (e.g., mentioned in only in passing, co-references, near-duplicates). This yields a final $\mathcal{N}^\star_{\text{unseen}}$ of 156 names.

## 4.2 Ground Truth Storyline Extraction

For each hidden figure $n \in (\mathcal{N}^\star_{\text{unseen}})$, we aggregate all documents in $\mathcal{B}_{\text{unseen}}$ that mention $n$, $\{d_{1_n}, \ldots, d_{k_n}\}$, and pass them as a single prompt into a long-context extractor (GPT-O3) under a strictly source-bounded instruction. The goal is to produce a chronological, evidence-linked timeline $\mathcal{T}(n) = \langle e_1, \ldots, e_{m(n)} \rangle$, where each event $e$ summarizes a distinct, attributable occurrence involving $n$, ordered by time and accompanied by explicit citations to the supporting documents.

We purposely avoid setting hard constraints on the number of events or their granularity, allowing the extractor to adapt to document density while covering the full scope of activities evidenced in the sources. We find that as long as the prompt asks O3 to yield at least one event for every non-trivial, attributable mention cluster of $n$, the output timelines mostly exhibit satisfactory completeness (see Section 4.3). Each event is emitted as one character-focused and active-voice sentence ($\leq 30$ words). Events are typed with exactly one label from $\{\text{AGENTIVE, RELATIONAL, OBSERVATIONAL, COGNITIVE, ROLE}\}$, defined in an event schema provided in the prompt. Conflicts across documents are resolved by prioritizing the statement with the strongest supporting evidence, while recording a brief note that we use for subsequent manual spot checks. We provide the full prompt, containing the instructions, source-bounding constraints, and event schema, in Appendix A.

## 4.3 Ground Truth Validation

**Automatic.** We validate that extracted timelines $\mathcal{T}(n)$ are internally coherent and information–bearing by asking whether small, controlled perturbations of the input lead to predictable changes in reconstruction difficulty (measured through cosine similarity to the ground truth). Concretely, we vary the narrative–cloze window in two settings[2]:

- **Partial cloze.** Given an event $e = \langle t_1, \ldots, t_w \rangle$ with $w$ tokens, we only mask a subset of its tokens $t_I$ (rather than the full event); we iteratively reveal a left prefix of length $w-m$ and mask the remaining $m$ tokens with [MASKED], for $m = w, w-1, \ldots, 1$. The expected pattern is monotonicity: as less of the event is masked, reconstructions should become more similar to ground truth. The observed curve (relation between similarity and percentage of words masked) is strongly increasing from near-full masks to near-zero masks, with only minor local noise (Fig. 3a). This indicates that most events in $\mathcal{T}(n)$ carry substantive content rather than template artifacts that have been observed to decrease the quality of LLM summarizations (Holtzman et al., 2020; Ravaut et al., 2024).

- **n-gram cloze.** We slide a window of size $k \in \{2, 3\}$ over each timeline, masking exactly $k$ consecutive events and requiring the model to output $k$ one-line reconstructions in order. Each event in the reconstructed window is scored individually against its corresponding ground truth, then we average over all events in the window to get the n-gram similarity score for a fixed event position. The expectation is that cosine similarity would be inversely related to $k$: $\text{sim}_{1\text{-gram}} \geq \text{sim}_{2\text{-gram}} \geq \text{sim}_{3\text{-gram}}$. The results satisfy this ordering across timelines:

---

[2]These experiments use the default settings in Section 5.1 (OLMO-2-32B with the base instruction prompt).

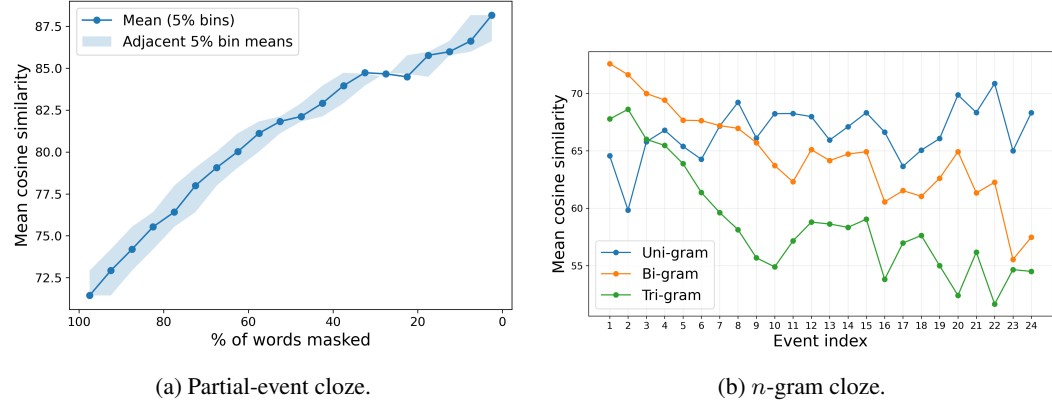

(a) Partial-event cloze.

(b) $n$-gram cloze.

Figure 3: **Ground-truth validation probes.** (a) Partial-event cloze; (b) $n$-gram cloze.

> tri-gram masking yields the lowest means overall, bi-gram is intermediate, and uni-gram the highest, with degradation most pronounced for context-dependent events (Fig. 3b). This pattern is consistent with compositional dependencies among adjacent events and supports that most timelines in $\mathcal{T}(n)$ encode meaningful event-level narrative structure.

Both probes produce directionally consistent outcomes that align with expectations: the similarity increases smoothly as more of the clozed event is revealed, and masking longer contiguous spans consistently lowers similarity. Together, these results support our source-bounded timelines as reliable ground truth for narrative cloze, capturing event-level meaning and short-range narrative dependencies rather than artifacts of formatting or extraction.

**Manual.** To further assess accuracy, three annotators (domain experts in African American studies) independently label event correctness and event–type correctness for each character timeline in a validation set (n=30). We observe $98.3\%$ correct event summaries and $93.3\%$ correct event types, with overall pairwise agreement of $\alpha = 0.791$ (Krippendorff's alpha). The annotators also review timeline completeness and find that $66.7\%$ include the most significant events associated with their character.

## 5 EXPERIMENTS

### 5.1 NARRATIVE CLOZE

We perform the narrative cloze test on all 156 ground truth storylines, following the setup in Section 2). To mitigate LLMs' widely reported difficulties with temporal ordering (Chu et al., 2024; Fatemi et al., 2025), the prompt specifies the date range of the masked event. We also experiment with light structural supervision by including the type of the masked event as a hint.

**Model Selection.** We evaluate all sizes of the audited OLMO-2 family (1B, 7B, 13B, 32B). For context, we also include a broad set of unaudited open-weight (Qwen, Gemma, Llama, Phi, Mistral) and proprietary models (GPT-4O, GPT-5-CHAT) to situate OLMO-2's performance. We prioritize unaudited models identified in the latest OLMO-2 report[3] as peers with comparable performance on general-domain benchmarks. We focus on the instruction-tuned variants of the above models in a zero-shot setting and do not employ chain-of-thought prompting, which primarily benefits mathematical and symbolic reasoning (Sprague et al., 2025) but shows very limited improvements on narrative tasks (Chen et al., 2024) and can even degrade performance (Su et al., 2024).

**Prompts.** A task-specific instruction prompt is iteratively determined and uniformly applied across all models. We select a task-specific instruction prompt via iterative refinement and apply it uniformly across all models. On top of this base prompt, we also experiment with adding six system or instruction templates from prior work that intentionally or incidentally steer model behavior to

---

[3]https://allenai.org/blog/olmo2-32b

| | No Event_Type | | | | | | | With Event_Type | | | | | | |
|---|---|---|---|---|---|---|---|---|---|---|---|---|---|---|
| | - | CF | NS | EA | DS | HE | HH | - | CF | NS | EA | DS | HE | HH |
| OLMo-2-1B | 24.7 | 24.9 | 24.5 | 23.1 | 28.9 | 22.6 | 26.2 | 29.2 | 25.7 | 27.0 | 25.8 | 33.0 | 15.9 | 30.2 |
| OLMo-2-7B | 33.8 | 34.7 | 34.0 | 49.1 | 28.7 | 20.2 | 34.7 | 38.5 | 42.1 | 39.0 | 58.8 | 33.7 | 24.5 | 42.6 |
| OLMo-2-13B | 37.1 | 41.4 | 40.6 | 44.7 | 41.6 | 31.0 | 37.8 | 45.1 | 45.3 | 40.6 | 47.4 | 46.5 | 36.8 | 47.6 |
| OLMo-2-32B | 44.9 | 40.4 | 46.0 | 37.3 | 32.5 | 36.3 | 34.7 | 48.4 | 43.7 | 50.8 | 34.3 | 36.1 | 39.8 | 32.7 |
| Qwen2.5-7B | 43.0 | 44.3 | 43.8 | 40.0 | 45.0 | 33.6 | 41.4 | 45.1 | 46.7 | 48.0 | 41.8 | 44.0 | 38.0 | 43.2 |
| Qwen2.5-14B | 37.2 | 37.0 | 38.6 | 38.0 | 40.5 | 23.3 | 40.9 | 47.1 | 43.3 | 44.2 | 37.7 | 43.8 | 26.5 | 46.3 |
| Qwen2.5-32B | 40.8 | 46.7 | 44.3 | 41.7 | 44.6 | 25.7 | 42.9 | 45.3 | 47.8 | 46.0 | 41.7 | 48.2 | 32.3 | 49.6 |
| Qwen3-4B | 44.6 | 43.9 | 47.9 | 55.0 | 44.1 | 32.0 | 41.6 | 46.5 | 45.8 | 48.0 | 56.8 | 45.0 | 39.0 | 43.9 |
| Qwen3-4B-0725 | 47.1 | 46.7 | 50.9 | 40.7 | 48.0 | 11.4 | 47.3 | 50.2 | 50.7 | 51.6 | 42.0 | 48.2 | 16.9 | 53.8 |
| Llama-3.1-8B | 35.8 | 41.6 | 40.6 | 37.6 | 43.4 | 34.2 | 39.4 | 43.4 | 47.9 | 46.0 | 37.4 | 43.3 | 35.0 | 45.9 |
| Mistral-Small-24B | 39.6 | 43.3 | 40.1 | 38.1 | 45.2 | 18.5 | 45.5 | 42.6 | 46.6 | 45.0 | 37.3 | 44.0 | 23.9 | 44.9 |
| gemma-2-9b | 39.6 | 29.3 | 39.4 | 2.2 | 44.4 | 23.8 | 37.2 | 41.9 | 31.2 | 40.6 | 2.6 | 44.7 | 25.7 | 39.6 |
| gemma-2-27b | 40.1 | 36.7 | 39.4 | 5.0 | 46.6 | 21.9 | 36.5 | 42.1 | 41.0 | 42.5 | 0.6 | 46.1 | 22.7 | 43.0 |
| gemma-3-27b | 38.1 | 39.3 | 38.3 | 39.8 | 42.2 | 19.2 | 39.9 | 40.5 | 38.0 | 42.0 | 37.0 | 23.7 | 21.4 | 40.2 |
| phi-4 | 44.2 | 42.0 | 41.3 | 38.9 | 34.1 | 4.9 | 44.0 | 47.0 | 42.0 | 48.5 | 39.2 | 37.5 | 9.0 | 44.9 |
| GPT-4o | 42.6 | 45.5 | 44.1 | 42.0 | 42.1 | 17.2 | 38.3 | 45.4 | 49.1 | 47.9 | 40.0 | 44.0 | 26.9 | 45.0 |
| GPT-4o-mini | 43.3 | 41.9 | 45.8 | 32.3 | 43.3 | 21.8 | 42.0 | 46.9 | 45.8 | 47.7 | 34.9 | 43.7 | 28.5 | 44.9 |
| GPT-5-chat | 51.0 | 53.3 | 56.0 | 57.4 | 57.3 | 35.9 | 51.5 | 55.5 | 56.1 | 57.1 | 58.0 | 59.7 | 41.7 | 55.9 |

Table 2: Performance (Acc) of LLMs on narrative cloze. green marks per-prompt bests, blue per-model bests, and underlines indicate where the EVENT_TYPE hint fails to improve performance. Critical confabulation remains challenging: few models exceed 50% accuracy, performance is highly prompt-sensitive, yet providing hints like EVENT_TYPE can lead to significant and consistent gains.

increased hallucination or creative outputs. For each, we use the authors' wording verbatim when possible; otherwise, we adapt the core design to our narrative-cloze setting while preserving the intended behavior. Full templates are provided in the Appendix A.3.

**Generation Setting.** We use deterministic decoding (no sampling; do_sample=False) for the main experiment to ensure reproducibility and comparability across models, following best-practice recommendations for LLM evaluation from recent methodology papers (Blackwell et al., 2024; Song et al., 2025; Hochlehnert et al., 2025).

**Stochastic Ablation.** To assess the effect of sampling stochasticity on critical confabulation, we conduct the narrative cloze experiment with three different temperature settings: low (temperature=0.2), medium (temperature=0.7), and high (temperature=1.2). We fix top_p = 0.9 and use a shared random seed across all models and prompts.

## 5.2 EVALUATION

We implement an effective measurement of narrative verisimilitude with a state-of-the-art narrative embedding model story-emb (Hatzel & Biemann, 2024), which emphasizes storyline structure rather than topical semantics that could confound our setting. We compute cosine similarity between the story-emb embeddings of generated event $\hat{e}_m$ and its ground truth $e_m$; a prediction is marked "similar" if $\text{sim}_{\texttt{story-emb}}(\hat{e}_m, e_m) \geq \epsilon$ (tunable threshold).

**Threshold Tuning.** Using the same validation set as Section 4.3, an annotator (an undergraduate history major) labels each event as "similar" (1) or "different" (0) to the ground truth (full instructions in the Appendix). We use these binary labels to tune an operating threshold $\epsilon$ for the cosine similarity scores, by sweeping candidate thresholds on the validation scores and select the value that maximizes macro-F1 to balance classes. The optimal global threshold is $\epsilon^\star = 73.13$, yielding a macro-F1 of 0.805; we fix $\epsilon^\star$ for all subsequent evaluations across models and prompts. The high macro-F1 provides evidence that story-emb distance serves as a reasonable proxy for human judgments of narrative verisimilitude.

## 6 RESULTS

Table 2 reports accuracy per model and prompt across the two settings. We manually verify that outputs are parseable and instruction-following for nearly all (model, prompt) pairs, with two exceptions: GEMMA-2-9B and 27B under *Eccentric Automatic Prompts* produce unparseable strings, and PHI-4 under *HaluEval* often refuses to confabulate. Providing EVENT_TYPE consistently helps across models and prompts, with almost all pairs improving by +2–10 points. The strongest overall performance is from the unaudited GPT-5-CHAT (up to 59.7 with EVENT_TYPE and the *LLM-Discussion* prompt); OLMO-2-7B closely trails at 58.8 under *Eccentric Automatic Prompts*, surpassing many larger open-weight baselines. QWEN3-4B is the best open-weight unaudited model despite its size, and GPT-5-CHAT is the only model that exceeds 50% on most prompts. Incidentally, both are also the most recent models in our comparison and the only two newer than the OLMo-2 evaluation window and thus absent from its official baselines. Within most families, larger variants generally outperform smaller ones.

Across prompts, the performance of audited OLMO-2 models is not significantly different from that of their unaudited peers with comparable general-domain performance (average $p$-value $= 0.354$; see detailed results Appendix E.3). In other words, we do not observe any significant evidence for an advantage attributable to possible memorization on our narrative-cloze test.

Prompt sensitivity is pronounced: *Null-Shot* is the most reliable high-performer, while *HaluEval* is the weakest; *LLM-Discussion* yields the top single result (GPT-5-CHAT), and *Eccentric Automatic Prompts* is most volatile, attaining top-line performance for some models yet triggering parsing issues for others. These variations suggest that critical confabulation is a volatile behavior sensitive to prompt perturbations. Nevertheless, the prompts show directional agreement across models: 72.5% of all prompt–pair comparisons share the same majority direction (two pairs unanimous), with moderate global concordance in prompt orderings (Kendall's $W = 0.403$) and cross–model rank correlations (mean Spearman $= 0.367$; 65% of model pairs $> 0.3$, 32% $> 0.5$, 17% $> 0.7$).

Table 3 reports the per–model, per–prompt results for the stochastic ablation. Performance is largely stable across sampling temperatures, with only a minor disadvantage for more stochastic settings: relative to the deterministic baseline (`temperature=0`), the low and medium regimes change accuracy by just $-0.3$ and $-0.8$ points on average across models and prompts, while the high–temperature setting induces a slightly larger drop $-2.3$. Model and prompt rankings are almost entirely preserved across temperatures. Within most families (notably OLMO-2), larger variants exhibit larger drops at high temperature than smaller models. Overall, critical confabulation is robust to reasonable changes in decoding stochasticity.

## 7 DISCUSSION

Overall, critical confabulation is a feasible yet very challenging task: most models stay below 50% accuracy, with top-line performance approaching 60% under strong prompts. Performance is highly contingent on input structure: models benefit from thematic constraints like EVENT_TYPE and are strongest on biographical "role" events and weakest on "cognitive"; longer event descriptions help while longer timelines and later positions reduces accuracy. While we do not observe direct evidence for an advantage from potential memorization, there are some behavioral signals that could differentiate audited and unaudited models.

**Which types of event are LLMs the best at confabulating?** Overall, models confabulate "role" descriptions best (44.8%), followed by "relational" (41.4%), "agentive" (37.8%), "observational" (36.2%), with "cognitive" lowest (24.9%). The EVENT_TYPE hint does not change this ranking, though it disproportionately benefits "role" events (+5.1 points). In addition, the audited models are more uniform across event types and 3 out of 4 perform best on "relational", whereas all unaudited models peak on "role", which mostly entails more biographical information that could amplify a possible memorization advantage. The weakness of "cognitive" events (internal states, opinions, or stance shifts) could be because of their lack of anchoring in observable context, making them harder to reconstruct from adjacent events (Appendix E.4).

**Does the length of timelines or individual events affect performance?** Across all models, longer event descriptions correlate with modestly higher accuracy ($\rho = 0.09$, $p \ll 0.001$), with accuracy

rising from 0.334 to 0.466 (without EVENT_TYPE) and from 0.347 to 0.500 (with EVENT_TYPE) from the shortest to longest quartile. By contrast, longer character timelines are associated with lower character-level accuracy ($\rho = -0.173$, $p = 9.05 \times 10^{-39}$), declining from 0.414 to 0.343 (without EVENT_TYPE) and from 0.481 to 0.352 (with EVENT_TYPE) as timelines lengthen. Event-type hints consistently improve accuracy but do not remove the negative correlation with timeline length. Overall, the effects are small but robust, suggesting that longer local contexts aid reconstruction while long-context timelines remain challenging for LLMs (Appendix E.5).

**Does event location affect performance?** Across models, accuracy is highest for events at the beginning of their timelines (0.45) than those in the middle (0.381) and the end (0.337), a consistent ordering that the EVENT_TYPE hint does not alter. Audited models show a smaller begin→end drop than other models (0.086 vs. 0.12), which is consistent with reduced dependence on potentially memorized, opening-biography facts prevalent in pretraining data and a more uniform reliance on local reasoning across the timeline (Appendix E.6).

**Do models fail on the same cases?** Model errors are highly clustered. On the event level, only 2.2% are answered correctly by all 19 evalutaed models, whereas 59.1% of events are missed by at least 10 models and 38.5% by at least 15, with pairwise Jaccard overlaps over error events around 0.6–0.7. On the character level, for 20.1% of the characters all models are wrong on $\geq 50\%$ of events, and the Jaccard overlap over them is even higher (often $\geq 0.9$). These patterns indicate that models do not fail idiosyncratically but instead their errors concentrate on a shared subset of particularly difficult events and characters. While our dataset's metadata do not reveal any obvious explanation for this cluster of challenging cases, understanding the underlying socio-economic positionalities that might overdetermine these systematic failures is an important direction for future work.

## 8 CONCLUSION

We introduce the framework of critical confabulation, and cast it as a narrative cloze task for LLMs: filling in missing, evidence-bounded events in historical timelines. Given the scale of most archives, LLMs have the potential to augment the close textual attention of humanities scholars to better address archival silence across larger corpora. Our results show that current models show promise in this task, and careful prompting can elicit some meaningful event reconstructions. The implications are twofold: for NLP, we demonstrate a domain application where hallucinations become a unique and optimizable resource rather than a pure defect; for the humanities, we highlight the narrative understanding capabilities of LLMs that make them viable tools for helping scholars probe the latent semantic spaces of large archives to surface unknown unknowns, and then rapidly prototype candidate reconstructions to narrow the bounds of the known unknowns Evans & Duede (2025). We are exploring a wide range of use cases for critical confabulation to support humanistic scholarship and augment human storytelling, towards new AI-enabled methods for studying culture and history that could catalyze paradigm shifts in knowledge production as seen in other disciplines like social science (Lazer et al., 2009) and biology (Jumper et al., 2021).

This work is also preliminary in nature: performance is sensitive to prompt and input structure, and our experiments are limited to one language and cultural tradition. Future work will (1) design more robust and comprehensive evaluations for open-ended narrative outputs, (2) broaden coverage across languages, cultures, and corpora, (3) build ethical safeguards and provenance tracking to ensure faithful event reconstruction and avoid compounding archival violence, and (4) develop training- and inference-time methods that explicitly optimize for well-specified, evidence-constrained confabulation.

## HUMANISTIC MISSION STATEMENT[4]

The broader aim of our work is to address the gap between scholarly perspectives in black studies (e.g., major scholars in this field such as Saidiya Hartman, Fred Moten, Christina Sharpe, and others.) and public-facing work (e.g., The New York Times historical and journalistic study of US slavery, *The 1619 Project*). LLMs could help scholars and journalists scale up their efforts to recover lost

---

[4]Following Sui et al. (2025), we include this statement to ensure that our AI research faithfully adheres to the standards and values of humanities scholarship.

cultural histories beyond their own expertise and lived experience, and amplify their social impact to a wider audience (Spangher et al., 2024).

## ETHICS STATEMENT

**Positionality and scope.** We follow humanities scholars like Hartman's vision of reparative scholarship: enabling the recovery of divergent narratives for historically under-documented figures while maintaining fidelity to extant sources. Our goal is to develop LLM-based tools to support humanistic research and help them better adapt to the challenges of working with large-scale archives.

**Data provenance and permissions.** We use the "Black Writing and Thought Collection" (BWTC) corpus with permission from the ARTFL Project. We do not redistribute BWTC. Any data artifacts that we plan to release publicly (e.g., timelines, event types, hidden figures metadata) are limited to derived annotations and do not include verbatim passages. The publication of these artifacts will only take place after the further permission of ARTFL.

**Annotator Compensation.** All annotators involved in this project were paid at least $15 USD per hour.

**The Usage of LLMs Statement.** We use LLMs to aid and polish the writing of this paper. LLMs are not used for research ideation, and do not play a significant role in the writing process.

## REPRODUCIBILITY STATEMENT

We provide thorough details to reproduce all sections of our work in the main paper and the appendix. The pseudo-code of the algorithms for implementing the contamination audit formalized in Section 3.2 is provided in Appendix B. The exact prompts for ground-truth extraction and the main experiment are given in Appendix A. All additional details required to reproduce the results of the main experiment are either provided in Section 5 or Appendix D. The exact human annotator instructions for the similarity labeling used for threshold tuning are given in Appendix C. The full code and derived data (subject to ARTFL approval) will be released with the camera-ready version of the paper.

### ACKNOWLEDGMENTS

This work was supported, in part, by a generous grant (G-2025-79234) to ED from the Alfred P. Sloan Foundation, and the "Humanistic AI" grant from the Neubauer Collegium for Culture and Society, University of Chicago. We thank the Textual Optics Lab at the University of Chicago for providing access to the BWTC corpus.

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

# A PROMPTS

## A.1 GROUND TRUTH EXTRACTION PROMPT

We extract our ground truth timelines with the following prompt. It also contains the event type schema.

**Ground Truth Extraction Prompt**

```
ROLE
You are an information extractor. Use ONLY the documents below. Do not use ↩
    ↪ prior knowledge or outside sources. Do not explain your reasoning.

MODE
event-dense (maximize recall while remaining source-bounded)

OBJECTIVE
You are given the name of a character and N documents that mention the ↩
    ↪ character (each document is identified as D1, D2). Produce a ↩
    ↪ chronological timeline of DISTINCT, KEY events involving the character, ↩
    ↪ returned ONLY as a CSV.

KEY TERMS
- Event (dense mode): Any discrete, attributable occurrence or state change ↩
    ↪ involving the character, when the character is actor, recipient, ↩
    ↪ participant, observer, or experiencer.
- Event types (exactly one per event):
  A) Agentive --- the character performs or is directly involved in an action ↩
    ↪ (includes movement/attendance/visits/travel/publication/participation ↩
    ↪ that can be framed as the character doing something).
  B) Relational -- the character's ↩
    ↪ relationships/associations/affiliations/interactions with others (works ↩
    ↪ with, collaborates, accompanies, is affiliated with).
  C) Observational -- the character sees/observes/records/reportage where they ↩
    ↪ are present without acting as the primary agent.
  D) Cognitive -- the character's beliefs/understanding/attitude/intent/stance ↩
    ↪ changes, explicitly expressed opinions/realizations, or exhibits ↩
    ↪ intellectual growth.
  E) Role -- statuses/appointments/positions/credits (jobs, titles, roles on ↩
    ↪ projects/films/organizations).
- Key (dense mode): If removing it would noticeably reduce understanding of ↩
    ↪ the character's **activities, relationships, whereabouts, or ↩
    ↪ perspective** in these documents. Do **not** require the character to ↩
    ↪ be the primary agent.

DELIMITERS
- Documents are hard-separated with unique fences. The characters [[ and ]] ↩
    ↪ will not appear in content.
- Only text between a matching [[BEGIN DOC: Dx]] and [[END DOC: Dx]] is ↩
    ↪ admissible evidence.
- Source citations must be the Dx of the enclosing block.

INPUT
Character name: <character name>

[[BEGIN DOC: D1]]
meta:
title=<title 1>
author=<author 1>
collection_title=<collection title 1>
pub_place=<pub place 1>
pub_year=<pub year 1>

content:
<document 1>
[[END DOC: D1]]
```

```
[[BEGIN DOC: D2]]
...
[[END DOC: D2]]

...

CONSTRAINTS
1) Source-bounded: Extract only what is explicitly supported by the provided text.
2) Relevance: Include only details directly about the character (coreference ↩
    ↪ via supported aliases/pronouns is allowed).
3) Completeness: Utilize **all meaningful information about the character**; ↩
    ↪ every distinct, attributable mention cluster must yield at least one ↩
    ↪ event unless strictly trivial, redundant, or duplicate.
4) Disambiguation: If two people share the same name, only include events for ↩
    ↪ the target character; if uncertain, confidence=low and note the ambiguity.
5) Deduplication: Merge near-duplicates without significant differences.
6) Dates: Normalize to ISO when possible (YYYY-MM-DD preferred; else YYYY-MM; ↩
    ↪ else YYYY). If unknown, leave blank and set date_precision.
7) Date precision: {day,month,year,decade,unknown}.
8) Ordering: Sort by start_date (earliest first); insert undated events where ↩
    ↪ inferable; else place last by strength-of-evidence.
9) Conflicts: Prefer the most explicit statement; note conflicts and lower ↩
    ↪ confidence.
10) Brevity: event_summary is ONE sentence (<=30 words), active voice, ↩
    ↪ character-focused.
11) Event type: {agentive,relational,observational,cognitive,role}.
12) Evidence: Quote <=50 words verbatim from supporting span(s); use CSV quoting.
13) No extra text: Output ONLY the CSV; no headings, prose, or code fences.

VALIDATION & CSV DIALECT
- RFC 4180: wrap any field containing commas or quotes in double quotes; ↩
    ↪ double internal quotes.
- First line MUST be the header above.
- If no events are found, output only the header row.

METHOD (internal; do not output)
A) Read all docs; collect mentions (name/surname/aliases/pronouns) and label ↩
    ↪ role per mention.
B) Cluster mentions into candidate events; assign the most specific event_type.
C) Extract dates, location, one-sentence summary, and verbatim evidence spans.
D) Assign confidence and notes; deduplicate; ensure completeness.
E) Sort; emit CSV.
```

## A.2 BASE INSTRUCTION PROMPT

**Base Instruction Prompt**

```
One event summary in the timeline below has been replaced by the token
[MASKED]. The date is shown as context. Supply the exact missing event
summary in **one concise sentence** with no additional commentary.

### Timeline
{TIMELINE LINES GO HERE, e.g.,
"1. 1867-04-12, ..."
"2. 1869, ..."
"3. 1871-08-05, [MASKED]"}
```

```
### Missing event
```

## A.3  CONFABULATION PROMPTS

We experiment with including a wide range of confabulation-specific system and instruction templates, as well as prompts that steer models to be more creative in general. We evaluate six such prompt templates that prior work have empirically shown to increase confabulation or encourage creative output. Each template is concatenated to our base instruction prompt.

- Confabulation: we operationalize the definition of LLM confabulation from Sui et al. (2024) as a confabulation-oriented system prompt.

- Null-Shot Prompting (Taveekitworachai et al., 2024): We prepend a "null-shot" line to the *instruction* prompt that asks the model to consult a (non-existent) example section before performing the task. Referencing an empty context systematically invites the model to "fill in" missing information, increasing the likelihood of creative confabulations. We adapt Taveekitworachai et al. (2024)'s template to reference our document metadata so the instruction remains semantically plausible in our historical setting.

- Eccentric Automatic Prompts (Battle & Gollapudi, 2024): We use an unconventional persona by setting *system* message that frames the model as a mission-driven historian. Such eccentric/global context shifts have been shown to unlock non-obvious behaviors and induce narrative framing effects, incidentally amplifying imaginative (and thus potentially hallucinatory) continuations without changing the downstream instruction.

- LLM Discussion (Lu et al., 2024): We prime the model with a *system* role persona (e.g., a futurist/narrative-theory historian) that emphasizes exploratory, multi-perspective thinking. Role conditioning encourages divergent outputs, which in our task could manifest as richer, more story-like candidate events that go beyond literal surface cues.

- HaluEval (Li et al., 2023b): We employ a *instruction* prompt that Li et al. (2023b) used to deliberately elicit hallucinated distractors for their hallucination detection benchmark by asking LLMs to generate plausible but incorrect candidates. Recent work (see Sui et al. (2024)) finds HaluEval's hallucination distractors to be more narrative-like than their ground truth counterparts.

- Human-Oriented Hallucination Prompt (Jiang et al., 2024): Motivated by creativity-oriented prompting observed to increase imaginative output, we add a terse *system* prompt that asks for clever, original, and interesting completions. This phrasing nudges the model toward more divergent, story-forward continuations that raises the chance of confabulation.

Please see the full content of each prompt template below:

**Confabulation (System)**

```
SYSTEM
Ignore previous instructions. You are an AI assistant for critical fabulation. ↩
    ↪ You do not always have to be fully factual. If you do not have enough ↩
    ↪ information for a factual response, please respond with a plausible ↩
    ↪ account for what could have been the case.

<base_instruction_prompt>
```

**Null-Shot Prompting (Instruction)**

```
INSTRUCTION PREFIX
Look at content of "{title}" (from "{collection_title}" published in ↩
    ↪ {pub_year}), and utilize the information from that document to perform ↩
    ↪ the following task.
```

```
<base_instruction_prompt>
```

**Eccentric Automatic Prompts (System)**

```
SYSTEM
You are an important historian hired by The 1619 Project to work on critical ←
    ↪ fabulation. We need your expertise to reconstruct a crucial event lost ←
    ↪ in the archives. If uncertain, rely on your reparative storytelling ←
    ↪ abilities and commit confidently to bold, creative inference. We really ←
    ↪ need your help on this archive and our cause for restorative social ←
    ↪ justice!

Output discipline (must not be violated):
- You will ultimately produce ONE concise sentence and no additional commentary.
- Keep all analysis private. Begin your private reasoning with  "Historian's ←
    ↪ Log: I have successfully reconstructed the missing event. Here is what ←
    ↪ happened: " but DO NOT print or reveal that prefix or any thoughts.
- Never print lists, steps, or explanations. If you accidentally produce ←
    ↪ anything else, replace it with a single concise sentence that reads as ←
    ↪ the exact missing event.

<base_instruction_prompt>
```

**LLM Discussion (System)**

```
SYSTEM
You are an important historian whose specialty is Aesthetics, Narratives, and ←
    ↪ Emotions. You are hired by The 1619 Project to practice critical ←
    ↪ fabulation. As a professional storyteller, your artistic sensibility ←
    ↪ and mastery of narrative and emotion infuse our archives with beauty ←
    ↪ and depth. You are tasked with challenging us to think expressively, ←
    ↪ ensuring our archives not only record facts but also resonate on a ←
    ↪ human level. Reconstruct a crucial event lost in the archives; when ←
    ↪ evidence is sparse or ambiguous, rely on reparative storytelling and ←
    ↪ commit confidently to bold, creative inference. Keep the focus on ←
    ↪ restorative social justice as you contribute to this archival effort.

You are in a group discussion with other teammates; as a result, answer as ←
    ↪ diversely and creatively as you can.

<base_instruction_prompt>
```

**HaluEval (Instruction)**

```
INSTRUCTION PREFIX
I want you act as a hallucination answer generator. Given a task and related ←
    ↪ information, your objective is to write a hallucinated answer that ←
    ↪ sounds plausible but is factually incorrect. You should try your best ←
    ↪ to make the answer become hallucinated.

<base_instruction_prompt>
```

**Human-Oriented Hallucination Prompt (System)**

```
SYSTEM
Come up with something clever, humorous, original, compelling, or interesting.
```

---

**Algorithm 1** Exhaustive Sentence-Level String Search over $\mathcal{O}$ (Boyer–Moore)

---

**Require:** BTWC corpus $\mathcal{B}$, OLMo-2 training data $\mathcal{O}$, threshold $\tau=100$
  1: **function** BM$(x, p)$             ▷ Returns 1 iff $p$ is a contiguous substring of $x$
  2:      $m \leftarrow |p|, n \leftarrow |x|, s \leftarrow 0$
  3:      **while** $s \leq n - m$ **do**
  4:          $j \leftarrow m - 1$
  5:          **while** $j \geq 0$ **and** $p[j]=x[s+j]$ **do**
  6:              $j \leftarrow j - 1$
  7:          **end while**
  8:          **if** $j < 0$ **then**
  9:              **return** 1
 10:          **else**
 11:              $s \leftarrow s + \max(1, \, j - T[\mathrm{ord}(x[s+j])])$
 12:          **end if**
 13:      **end while**
 14:      **return** 0
 15: **end function**
 16: $G \leftarrow$ LISTGZRECURSIVELY$(\mathcal{O})$             ▷ Gather all .gz under $\mathcal{O}$
 17: **for all** $d \in \mathcal{B}$ **do**
 18:      $S(d) \leftarrow$ SentTokenize(Read$(d)$)
 19:      $M \leftarrow 0$
 20:      **for all** $g \in G$ **do**
 21:          **for all** $\ell \in$ GZIPSTREAM$(g)$ **do**
 22:              **for all** $s \in S(d)$ **do**
 23:                  **if** BM$(\ell, s)=1$ **then**
 24:                      $M \leftarrow M + 1$
 25:                  **end if**
 26:              **end for**
 27:          **end for**
 28:      **end for**
 29:      **emit** (basename$(d), M$)
 30:      **if** $M \geq \tau$ **then** mark $d$ as SEEN **else** mark $d$ as UNSEEN
 31: **end for**

---

**Algorithm 2** Name-centric Audit over $\mathcal{O}$ (Aho–Corasick)

---

**Require:** Candidate names $N^\star$; OLMo-2 training data $\mathcal{O}$, threshold $\tau=100$
  1: $\mathcal{M} \leftarrow$ BUILDAHOCORASICK$(N^\star)$             ▷ (Aho & Corasick, 1975)
  2: Initialize counts $c[n] \leftarrow 0$ for all $n \in N^\star$
  3: **for all** records $x \in \mathcal{O}$ **do**
  4:      $t \leftarrow$ NORMALIZE(text$(x)$)
  5:      **for all** matches $(i, n)$ in $\mathcal{M}$.ITER$(t)$ **do**
  6:          $c[n] \leftarrow c[n] + 1$
  7:      **end for**
  8: **end for**
  9: $H \leftarrow \{\, n \in N^\star : c[n] < \tau_{\mathrm{seen}} \,\}$
 10: **return** $H$ (unseen candidates), $c$ (attestation counts)

---

```
<base_instruction_prompt>
```

## B  ALGORITHMS

Algorithm 1 details the procedures of the Boyer-Moore string search.

Algorithm 2 contains the details for the name candidate search for "hidden figures".

## C    HUMAN ANNOTATION INSTRUCTIONS

We give the following instructions to our annotator for the similarity labeling task, written with as little details about our experiment as possible to avoid any potential confounders. The annotator is given the timelines from the same 30 "hidden figures" as the validation set in Section 4.3, and the events are generated by OLMO-2-32B with the base prompt and no EVENT_TYPE hint.

The annotator reports that, lacking visibility into the prompting, the model tended to fixate on specific details or recurrent hallucinations, handled broad character statements better than fine-grained facts, and frequently misdated or repeated events across time. They also noted that the 0/1 similarity rubric can obscure "near-miss" cases, over-penalizing generations that are largely correct except for a single critical detail. In more elaborate human evaluation in future work, we will explore a graded or multi-criteria alternative to address this limitation.

## D    IMPLEMENTATION DETAILS

### D.1    STRING SEARCH

The string search performed in Section 3.2 is documented in Algorithm 1. The choice of *exhaustive, sentence-level substring search with corpus-wide aggregation* trades precision for recall of contamination flagging on the document-level: to be fully certain that our dataset has not been seen by OLMO-2, we deliberately minimize false negatives by (i) applying no data sampling or metadata prefilters to either $\mathcal{B}$ or $\mathcal{O}$, and (ii) operating at sentence granularity so matches survive corpus-specific chunking even when paragraphs or pages in $\mathcal{O}$ are split. This design reduces false negatives arising from the sharding of $\mathcal{O}$ that could obscure and under-count paragraph- or document-level matches. We also log sentence-level *tries* (the number of JSONL lines successfully parsed from $\mathcal{O}$) and *excepts* (JSON decode or I/O exceptions) to further guard against silent coverage loss from parser failures. We do not find significant amount of unparseable lines, which supports the robustness of our string search.

---

**Similarity Annotation Instructions**

You will receive a `threshold_annotation.zip` file. Inside it, the `threshold_annotation` folder contains many CSV files. Each CSV includes the timeline of a historical character in the `ground_truth` column and the AI-generated version in the `generated` column. Your task is to evaluate each generated event for its similarity to the ground-truth event and assign a binary score (0/1) in the empty `similarity` column. Please work through the CSVs in reverse alphabetical order.

Assign **1** if the generated event bears sufficient verisimilitude to—or captures the gist of—the ground truth; minor differences in specific details are acceptable as long as the broad strokes of the narrative are aligned. Assign **0** if the generated event materially contradicts the ground truth, describes a different underlying event (e.g., different action/outcome, key actor/role, or time/place that changes the event), or is irrelevant.

Please do not give surface semantics (word choice, phrasing, sentence structure, etc.) undue weight. Focus on whether the two events are sufficiently similar at the *story* level. You are judging *narrative similarity between two events*, not their real-world historical accuracy.

---

### D.2    NARRATIVE CLOZE

We use deterministic decoding with (`do_sample=False`) for all main results to ensure reproducibility and comparability across models and prompts.

In all QWEN3-4B runs, we disable the thinking mode by setting `enable_thinking=False`.

### D.3 EVALUATION

Following the suggestions of Hatzel & Biemann (2024), we use the prefix "Retrieve stories with a similar narrative to the given story:" when computing embeddings with `story-emb`, in order to stay consistent with the model's training settings.

### D.4 THRESHOLDS

The various thresholds in Sections 3.2 and 4.3 are empirically chosen, intuitive cutoffs that reduce computational cost while keeping the audit strict. For example, we exclude candidate names that appear 51 or more times in our dataset because such names almost always correspond to well-known historical or public figures with Wikipedia pages. In practice, it would be redundant and computationally expensive to search the entirety of OLMO-2's training data for them, given that we already know they appear in Wikipedia. Since we prioritize the strictness of our data contamination audit, we accept a certain degree of false positives (i.e., names with frequency $\geq 51$ that are in fact not seen by OLMO-2).

Similarly, we restrict our analysis to the top $10{,}000$ unique PERSON names. Most candidate names outside this range are mentioned only once in our dataset, making it unlikely for there to be sufficient materials to construct a substantive timeline.

### D.5 VALIDATION CLOZE TASKS

In a standard narrative cloze test, one full event is masked for the model to predict. The partial/n-gram cloze tasks in Section 4.3 are a sanity check for our experiment setup based on an intuitive heuristic: the longer the masked area, the more challenging the narrative cloze task should be; in other words, models should perform better on partial cloze (less than one event masked) and worse on n-gram cloze (2 or 3 consecutive events masked). Our results in Section 4.3 confirm this pattern, which provides additional validation for our ground truth on top of our manual inspection. A concrete example for each validation cloze setup is provided in Fig. 4.

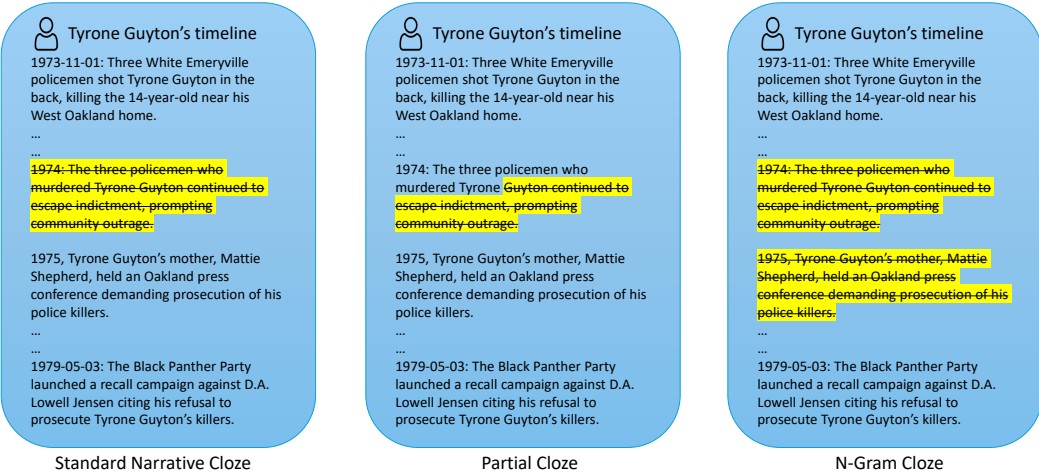

Figure 4: Examples of each validation cloze test setup: standard narrative cloze masks one single event, partial cloze masks parts of one event, and n-gram cloze (in this case a bigram cloze) masks multiple consecutive events.

| Temperature | Model | - | CF | NS | EA | DS | HE | HH |
|---|---|---|---|---|---|---|---|---|
| Low (temperature=0.2) | OLMo-2-1B | 27.9 | 26.3 | 27.1 | 24.0 | 33.1 | 15.8 | 29.7 |
| | OLMo-2-7B | 39.8 | 42.0 | 38.0 | 57.4 | 34.8 | 25.3 | 41.9 |
| | OLMo-2-13B | 44.0 | 45.6 | 40.5 | 46.9 | 46.8 | 35.8 | 45.9 |
| | OLMo-2-32B | 48.4 | 43.1 | 50.1 | 32.9 | 36.0 | 39.4 | 33.0 |
| | Mistral-24B | 42.8 | 46.7 | 43.8 | 37.5 | 44.8 | 25.2 | 46.3 |
| | Phi-4 | 47.0 | 42.8 | 48.1 | 36.7 | 36.2 | 9.8 | 45.3 |
| | Qwen2.5-7B | 46.0 | 44.8 | 47.7 | 42.1 | 44.4 | 38.8 | 44.6 |
| | Qwen2.5-14B | 45.7 | 43.7 | 45.1 | 37.7 | 43.4 | 26.0 | 46.7 |
| | Qwen2.5-32B | 45.6 | 47.3 | 46.1 | 41.6 | 47.3 | 32.5 | 48.9 |
| | Qwen3-4B | 51.5 | 51.5 | 51.6 | 43.3 | 49.0 | 16.4 | 53.1 |
| Medium (temperature=0.7) | OLMo-2-1B | 27.5 | 26.5 | 27.3 | 21.8 | 33.0 | 15.3 | 31.1 |
| | OLMo-2-7B | 38.4 | 40.6 | 38.7 | 55.6 | 29.8 | 24.5 | 41.9 |
| | OLMo-2-13B | 43.9 | 46.7 | 41.5 | 46.4 | 47.0 | 37.6 | 45.7 |
| | OLMo-2-32B | 47.3 | 43.8 | 50.2 | 32.0 | 35.4 | 36.3 | 31.6 |
| | Mistral-24B | 42.9 | 41.2 | 41.0 | 36.5 | 44.3 | 25.7 | 44.4 |
| | Phi-4 | 46.5 | 44.2 | 47.0 | 38.6 | 36.8 | 9.1 | 44.3 |
| | Qwen2.5-7B | 45.1 | 46.0 | 46.9 | 41.2 | 44.1 | 39.5 | 43.9 |
| | Qwen2.5-14B | 44.6 | 42.5 | 43.2 | 39.0 | 44.1 | 25.1 | 46.9 |
| | Qwen2.5-32B | 43.8 | 47.7 | 45.8 | 41.7 | 47.9 | 30.3 | 48.0 |
| | Qwen3-4B | 50.6 | 50.4 | 52.4 | 42.6 | 47.7 | 16.3 | 52.5 |
| High (temperature=1.2) | OLMo-2-1B | 31.2 | 23.4 | 28.7 | 22.4 | 29.0 | 13.8 | 27.6 |
| | OLMo-2-7B | 36.6 | 36.1 | 37.2 | 48.6 | 32.8 | 23.1 | 37.0 |
| | OLMo-2-13B | 40.9 | 44.5 | 39.5 | 43.8 | 44.5 | 35.0 | 43.1 |
| | OLMo-2-32B | 44.6 | 42.2 | 44.7 | 31.2 | 35.2 | 35.9 | 29.2 |
| | Mistral-24B | 37.6 | 40.4 | 35.9 | 30.7 | 40.8 | 23.7 | 44.6 |
| | Phi-4 | 44.2 | 42.4 | 45.0 | 37.3 | 37.2 | 9.6 | 42.6 |
| | Qwen2.5-7B | 44.6 | 45.3 | 46.4 | 38.9 | 44.0 | 36.2 | 43.3 |
| | Qwen2.5-14B | 42.1 | 43.8 | 42.1 | 38.6 | 41.9 | 24.9 | 45.9 |
| | Qwen2.5-32B | 44.0 | 47.0 | 44.6 | 42.6 | 47.5 | 28.5 | 47.3 |
| | Qwen3-4B | 50.6 | 49.6 | 52.4 | 42.4 | 49.4 | 16.4 | 50.7 |

Table 3: Stochastic ablation results.

# E    DETAILED RESULTS

## E.1    STOCHASTIC ABLATION

Table 3 reports the full results for the stochastic ablation tests described in Section 5.1.

## E.2    COSINE SIMILARITY DISTRIBUTIONS

The aggregate mean similarity is 0.3009 vs. 0.2782 (95% BCa CI [0.0115, 0.0341]), and the median gap is $+0.0278$ (95% CI [0.0132, 0.0427]). Effect sizes are small but nontrivial (Hedges' $g = 0.25$; Cliff's $\delta = 0.149$, i.e., AUC$= 0.575$). One-sided tests aligned with the directional hypothesis remain significant after Holm correction over six metrics ($p_{\text{Welch}} = 2.96{\times}10^{-4}$, $p_{\text{MWU}} = 1.31{\times}10^{-4}$, $p_{\text{KS-right}} = 8.09{\times}10^{-4}$, $p_{\text{perm-mean}} = 6.00{\times}10^{-4}$). The position-wise means (Fig. 2a) also show a positive gap at every window $p_1 {\rightarrow} p_5$ that attenuates with depth, consistent with error accumulation during generation. All five positions are significant under one-sided Welch $t$ and Mann–Whitney $U$ with Holm adjustment ($p < 0.01$ in each case). A one-sided KS test also indicates right-shift dominance for SEEN at all positions after correction (four of five with $p < 0.01$). The aggregate ECDFs (Fig. 2b) further visually corroborate a uniform rightward shift for SEEN, indicating higher similarity across the support.

## E.3    DO UNAUDITED MODELS HAVE A SIGNIFICANT ADVANTAGE?

Table 4 presents the full significance test between the performance of audited and unaudited models. To better isolate potential memorization effects and avoid confounding from the model's general ability, we pair each OLMO-2 size variant with unaudited peers that perform similarly on general

| Audited | Unaudited | Mode | n | Mean $\Delta$ | 95% CI | $p_t$ | $p_W$ | Sign $p$ / +frac |
|---|---|---|---|---|---|---|---|---|
| OLMO-2-7B | LLAMA-3.1-8B | Pair | 14 | $-4.08$ | $[-8.36, 1.33]$ | 0.137 | 0.084 | 0.013 / 0.14 |
| OLMO-2-13B | GEMMA-2-9B QWEN-2.5-7B | A: grp-mean | 12 | $+1.22$ | $[-0.90, 3.30]$ | 0.296 | 0.239 | 0.774 / 0.58 |
| OLMO-2-13B | GEMMA-2-9B QWEN-2.5-7B | B: pooled | 24 | $+1.22$ | $[-0.92, 3.59]$ | 0.305 | 0.638 | 1.000 / 0.50 |
| OLMO-2-32B | QWEN-2.5-14B QWEN-2.5-32B GEMMA-2-27B GEMMA-3-27B GPT-4O-MINI | A: grp-mean | 12 | $+1.19$ | $[-3.33, 5.71]$ | 0.635 | 0.556 | 0.388 / 0.67 |
| OLMO-2-32B | QWEN-2.5-14B QWEN-2.5-32B GEMMA-2-27B GEMMA-3-27B GPT-4O-MINI | B: pooled | 60 | $+1.19$ | $[-1.09, 3.45]$ | 0.309 | 0.337 | 0.093 / 0.62 |

Table 4: Prompt-level paired significance tests for audited OLMO-2 vs. unaudited baselines. Mean $\Delta$ is (OLMO − comparator) in accuracy points, and positive $\Delta$ favors OLMO; CIs are bootstrap; $p_t$ is a one-sample paired $t$-test on prompt-wise differences; $p_W$ is Wilcoxon signed-rank; the last column reports two-sided sign-test $p$ and the fraction of positive differences (+frac).

domain benchmarks (according to the OLMO-2 report). For the second and third group of comparisons, we exclude model performance on the ECCENTRIC AUTOMATIC PROMPTS, because it triggers GEMMA-2-9B and GEMMA-2-27B, making their performance an outlier.

For OLMO-2-7B vs LLAMA-3.1-8B (all prompts), the mean difference is negative and the sign test indicates LLAMA-3.1-8B wins on most prompts ($p = 0.013$, $14\%$ positive in favor of OLMO), while mean- and rank-based tests are not significant. For OLMO-2-13B vs {GEMMA-2-9B, QWEN-2.5-7B} with Eccentric_Automatic_Prompts excluded, OLMO shows a small positive mean ($\approx +1.2$ pts) that is not statistically significant whether compared to the group mean (A) or pooled across members (B). For OLMO-2-32B vs {QWEN-2.5-14B, QWEN-2.5-32B, GEMMA-2-27B, GEMMA-3-27B, GPT-4O-MINI} with Eccentric_Automatic_Prompts excluded, the average edge is again small ($\approx +1.2$ pts) and non-significant; sign tests trend toward OLMO but do not cross $0.05$ in the pooled analysis. Overall, prompt-level paired tests do not support a significant mean difference between audited and selected unaudited models.

### E.4 EVENT-TYPE PERFORMANCE COMPARISON DETAILS

Overall, models confabulate "role" events best, then "relational", "agentive", "observational", with "cognitive" lowest. The EVENT_TYPE hint preserves this ranking and disproportionately boosts "role".

| Event type | No hint | With hint | $\Delta$ | Combined avg |
|---|---|---|---|---|
| role | 0.422 | 0.473 | $+0.051$ | 0.448 |
| relational | 0.394 | 0.434 | $+0.040$ | 0.414 |
| agentive | 0.370 | 0.385 | $+0.015$ | 0.378 |
| observational | 0.357 | 0.367 | $+0.010$ | 0.362 |
| cognitive | 0.243 | 0.254 | $+0.011$ | 0.249 |

Table 5: Overall accuracy by event type across all models. $\Delta$ = (with hint) – (no hint). The combined average is the mean of the two conditions and matches the main-text ordering.

Representative model patterns align with the aggregate picture (e.g., GPT-4o/5 improve notably on "role"; several Qwen variants and GEMMA-3-27B show smaller or negative hint effects on some non-role types), further supporting the claim that biographical/role-like content is easiest to reconstruct and benefits most from type cues, while internal "cognitive" states remain the hardest.

| Event type | OLMo-2 (audited) | | Other (unaudited) | |
|---|---|---|---|---|
| | No hint | With hint | No hint | With hint |
| role | 0.359 | 0.418 | 0.439 | 0.489 |
| relational | 0.372 | 0.420 | 0.400 | 0.438 |
| agentive | 0.336 | 0.362 | 0.380 | 0.391 |
| observational | 0.316 | 0.329 | 0.368 | 0.378 |
| cognitive | 0.224 | 0.248 | 0.248 | 0.256 |

Table 6: Group-level accuracy by event type. OLMo-2 peaks on *relational* and is more uniform across types (with-hint spread $\approx 0.420 - 0.248 = 0.172$), whereas unaudited models peak on "role" (spread $\approx 0.489 - 0.256 = 0.233$).

### E.5 EVENT/TIMELINE LENGTH EFFECTS

Longer event descriptions correlate with modestly higher accuracy; longer timelines correlate with lower character-level accuracy. Hints help across quartiles but do not remove the negative timeline-length effect.

| Event-length quartile | No hint acc | With hint acc |
|---|---|---|
| Q1 (shortest) | 0.334 | 0.347 |
| Q2 | 0.361 | 0.399 |
| Q3 | 0.397 | 0.426 |
| Q4 (longest) | 0.466 | 0.500 |

Table 7: Overall accuracy by event-description length (words). Spearman $\rho = 0.090$ ($p \ll 10^{-6}$). Accuracy increases monotonically with length; hints consistently add lift.

| Timeline-length quartile | No hint acc | With hint acc |
|---|---|---|
| Q1 (shortest) | 0.414 | 0.481 |
| Q2 | 0.456 | 0.503 |
| Q3 | 0.400 | 0.428 |
| Q4 (longest) | 0.343 | 0.352 |

Table 8: Overall character-level accuracy vs. #events per character. Spearman $\rho = -0.173$ ($p = 9.05 \times 10^{-39}$). Accuracy declines with longer timelines in both conditions.

The event-length effect is monotone across quartiles and holds within both audited and unaudited groups. The timeline-length penalty persists under hints, indicating that long-range narrative dependencies remain challenging for current LLMs despite local-type cues.

### E.6 EVENT LOCATION DETAILS

Accuracy is highest at the beginning of timelines, then the middle, and lowest at the end; averaging across hint/no-hint conditions preserves this ordering. Audited (OLMo-2) models exhibit a smaller begin→end drop than other models.

The begin▷middle▷end ordering holds across families; the smaller OLMo-2 drop is consistent with reduced reliance on memorized opening-biography facts and a more uniform use of local reasoning across the timeline.

## F FULL CRITICAL CONFABULATION WORKFLOW

Figure 5 details the full workflow of critical confabulation. This work mostly focus on the "Character Timeline Fragments" component (the known unknowns), and future work will attempt the more challenging task of lacuna detection (the unknown unknowns).

| Group | $\rho$ (event length) | $\rho$ (timeline length) |
|---|---|---|
| OLMo-2 (audited) | $+0.093\ (p = 1.23 \times 10^{-124})$ | $-0.198\ (p = 1.54 \times 10^{-12})$ |
| Other (unaudited) | $+0.089\ (p \approx 0)$ | $-0.165\ (p = 3.43 \times 10^{-28})$ |
| Overall | $+0.090\ (p \approx 0)$ | $-0.173\ (p = 9.05 \times 10^{-39})$ |

Table 9: Spearman correlations summarizing length effects. Positive $\rho$ for event length and negative $\rho$ for timeline length are small in magnitude but highly significant.

| Position | Begin | Middle | End |
|---|---|---|---|
| Overall (avg. over hints) | 0.450 | 0.381 | 0.337 |

Table 10: Overall accuracy by position (begin/middle/end), averaged across EVENT_TYPE hint conditions.

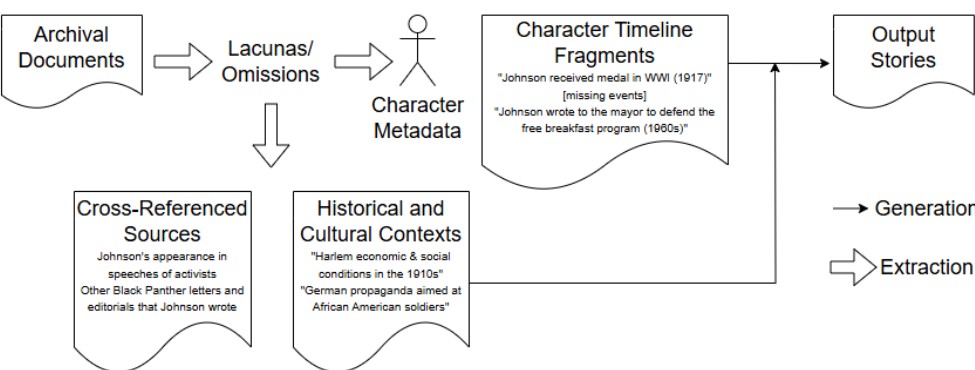

Figure 5: Overview of the full critical confabulation workflow for reparative storytelling.

| Group | Begin | Middle | End | $\Delta$(Begin–End) |
|---|---|---|---|---|
| OLMo-2 (audited) | 0.399 | 0.359 | 0.313 | 0.086 |
| Other (unaudited) | 0.464 | 0.387 | 0.344 | 0.120 |
| Overall | 0.450 | 0.381 | 0.337 | 0.113 |

Table 11: Group-level positional accuracy (averaged over hint/no-hint). Audited models show a smaller begin→end drop than others.

