# OpenReview forum: "Critical Confabulation: Can LLMs Hallucinate for Social Good?"
_ICLR.cc/2026/Conference — ICLR 2026 Poster_

### Official Review · Reviewer_ubaQ · 2025-10-31

**Soundness:** 4
**Presentation:** 3
**Contribution:** 3
**Rating:** 8
**Confidence:** 2

**Summary:**

The paper presents an LLM-based methodology for inferring historical narratives, given some historical grounding information. First, the paper establishes the epistemological roots of critical fabulation to situate the utility of their method. The paper then details the dataset used and how that dataset was cleaned to prevent contamination of the data with the model’s training data. Then, the paper tests how well LLMs can perform a confabulation task of reconstructing narratives. The paper finds that while LLMs are performant in this task, there is still a lot of room for improvement in future work.

**Strengths:**

The work is very novel and has very good quality in its empirical grounding. For novelty, the paper applies an LLM to a novel domain, namely the study of history. I am not aware of many applications of AI or machine learning to this domain. Additionally, there are also few works that exploit the hallucination/confabulation ability for academic pursuits. Many academic pursuits are based in facts and substantiated claims, so the use of the creative part of an LLM in the academic domain is both counterintuitive and novel.

For the quality of the paper, the data cleaning procedures are very thorough and detailed. The paper does a good job of controlling for things like data set contamination. Additionally, the paper also considers several variations in prompts and masking of events to fully understand the performance contours of LLMs for the narrative closure task.

**Weaknesses:**

The paper could benefit from some clarity enhancement. For example, I have some trouble picturing the differences between the validation cloze tasks. I think paper would benefit from providing some concrete examples of full timeline $\rightarrow$ partial cloze set up $\rightarrow$ n-gram cloze set up (maybe in an appendix). Additionally, it's not clear why some of the thresholds are what they are. For example, when extracting name candidates, why do you take the top 10,000 persons, and why is less than 51 the frequency to keep? Outside of some clarity enhancements that could be made, I think the method would be better if there were some sense of certainty in the predicted events. As seen in this research, this is a challenging task. And, previous research has found humans tend to overestimate AI’s capabilities. So, when these two things combine for a historical task where the confabulations are attempting to fill in for historical truth, I worry the method could result in bad or even harmful confabulations that could distort the historical picture for humans more than the current, partial view of history for unseen peoples and classes of people. Thus, I would surmise that having some estimates of confidence could help humans to better judge whether the confabulations are good or not.

**Questions:**

1. For curating the list of unseen names, how did you disambiguate between different personalities with the same name?

2. Where do the event types come from?

3. How did you deal with differences in time scales within a narrative trajectory (e.g., some events might tightly cluster together with long gaps of time between event clusters)?

---

> ### Author Response · Authors · 2025-11-24
> **Response to Weaknesses**
>
> We are pleased to learn that the reviewer appreciates the empirical rigor of our work, and recognizes the novelty of applying LLM to historical analysis and our attempt to leverage the positive affordances of hallucinations/confabulations that fills a wider gap in NLP (as was our hope).
>
> W1 (“The paper…”): We thank the reviewer for pointing out this issue. We have simplified the description of the validation cloze tasks in Section 4.3 for better clarity, and provided an example of each setup in the Appendix. In a standard narrative cloze test, one full event is masked for the model to predict. The partial/n-gram cloze tasks are a sanity check for our experiment setup based on an intuitive heuristic: the longer the masked area, the more challenging the narrative cloze task; in other words, models should perform better on partial cloze (less than one event masked) and worse on n-gram cloze (2 or 3 consecutive events masked). Our results (Section 4.3) confirm this pattern, which provides additional validation for our ground truth on top of our manual inspection.
>
> W2 (“Additionally…”): The thresholds are empirically determined to reduce computational cost. For instance, we exclude name candidates with 51 or more mentions in our dataset because most of them are famous historical figures with wikipedia pages, and it would be a waste of resources to search for them in OLMo-2’s training data as we already know that they are in Wikipedia. Since we intend our data contamination audit to be strict (we are glad that the reviewer appreciates our thoroughness), we accept a certain degree of false positives (names >= 51 frequency that are actually not seen by OMLo-2).
>
> We only take the top 10,000 persons because most named candidates outside of that range are only mentioned once in our dataset, which makes it highly unlikely that there would be enough material to construct a timeline.
>
> We added both clarifications as a new subsection, “Thresholds,” in the “Implementation Details” section of our Appendix.
>
> W3 (“Outside…”): The reviewer’s instincts are spot on. We agree with the reviewer’s concerns. Indeed, the risks associated with confabulation in historical tasks are, in many ways, distinct from those arising in confabulation tasks involving queries seeking facts that ought to be contained in a model’s training data. As we note, we see critical confabulation as part of the expert historian’s tool-kit and not as a stand-alone technology. That is, in agreement with the reviewer and consistent with our objectives, we believe that critical confabulation must be conducted in a human-in-the-loop setting. While this is, as it were, the north star guiding our work, the focus (and contribution) of this initial study is to lay the groundwork for precisely this kind of research (e.g., expert-mediated AI critical confabulation for research). Our study reveals the need for a human in the loop and what that human should be doing.
>
> Importantly, our results also demonstrate the feasibility of this approach. Certainty in tasks of this kind is hard-won, which is precisely why expert participation is indispensable. It is the integration of expert judgment with AI-driven critical confabulation that allows us to approach historical truth in ways that would otherwise remain out of reach. We have updated the manuscript’s discussion to clarify and reflect this.
>
> We thank you for this clarifying question!

---

> ### Author Response · Authors · 2025-11-24
> **Response to Questions**
>
> Q1: We conducted manual filtering (described at the end of Section 4.1) to remove co-references and near-duplicates. We read through each document the name appeared in to make sure they are referring to the same person, and we have not encountered a case where one name refers to two different personalities.
>
> Q2: We thank the reviewer for noting this point of potential confusion, and have edited our description (Section 4.2) for better clarity. The event types were annotated by o3 when constructing the ground truth timelines (Section 4.2 and full prompt in Appendix A). Our manual inspection finds that these event type annotations are generally reliable (93.3% accuracy).
>
> Q3: We thank the reviewer for highlighting a very interesting research question! In this study, we did not attempt to grapple with the influence of time, particularly within narrative trajectories. However, this is an important idea. Narrative trajectories often span long intervals precisely because key events occur outside the boundaries of the narrative itself. Part of what would be required to complete or extend such trajectories responsibly is some principled way of bringing those extra-narrative developments back into view.
>
> This, in turn, suggests a productive direction for future work. One could imagine supplementing the critical confabulation process with time-indexed “known knowns” drawn from the broader historical record, enabling the model to anchor its inferences to events that fall outside the textual frame but shape its internal logic. We gesture toward this possibility in our discussion of future research, and we see it as a natural next step in building more temporally robust critical confabulation systems.

---

### Official Review · Reviewer_kuNP · 2025-11-01

**Soundness:** 4
**Presentation:** 2
**Contribution:** 4
**Rating:** 6
**Confidence:** 3

**Summary:**

The authors propose critical confabulation: using constrained LLM “confabulations” to fill archival gaps in ways informed by humanities practice (Hartman’s critical fabulation). They operationalize this as a narrative cloze task over a Black-writing archival collection (BWTC): mask events in a character timeline and ask models to generate plausible evidence-bound fill-ins. They evaluate a range of audited open models (OLMo-2 family) and baselines (Gemma, GPT-4o, etc.) under prompting conditions intended to elicit controlled hallucination. Automatic metrics (cosine similarity, retrieval overlap etc.) and some qualitative examples are used to argue that constrained confabulation can produce plausible, useful candidate narratives.

**Strengths:**

I think this paper raises a very interesting and socially meaningful problem, and I love to see how people connect LLM capabilities to social good in unexpected ways. I think in order to do that, the authors have to really make sure that the task they choose for hallucinations should not be contaminated, and much of the heavy lifting in this paper is done at finding that task and data, and the authors have accomplished with careful justifications of data contamination (Section 3.2). I also love that the authors have spent great efforts in curating the dataset and diverse ablations with different prompting strategies. I also like that the paper even tests and analyzes settings between unaudited models.

**Weaknesses:**

I think the paper would benefit from more qualitative examples and analyses, like showing the failure modes of LLMs and try to analyze the hard data where most models are unable to get them right. In addition, the presentation can be improved. For instance, many figures in the paper have really small fonts and are very hard to see. Figure 1 can also be more dense and displayed with larger font, as I have to zoom in on a laptop to figure out what is being displayed.

**Questions:**

Have you thought about changing the temperature of LLMs to nonzero? Can you set seeds for the LLMs to still control reproducibility?

---

> ### Author Response · Authors · 2025-11-24
> **Response to Weaknesses and Questions**
>
> We are glad that the reviewer shares our interest in LLMs’ unexpected affordances for social good, and we are heartened that the reviewer sees the value of our careful work on the data contamination audit.
>
> W1 (“I think…”): We thank the reviewer for suggesting two questions that strengthen our qualitative analysis in Section 7, where we discuss the various qualitative factors affecting model performance. We have added an error taxonomy of common failure modes (with examples) and an error distribution analysis that shows LLMs do/don’t perform poorly on the same events. We have also made changes to the narrative, where appropriate, to draw the reader’s attention to these qualitative examples and their importance. We appreciate this suggestion as we feel that it helps to focus on what is the most interesting about the paper.
>
> W2 (“In addition…”): We regret that the presentation of our manuscript is limited by the small fonts of our figures. We have updated all the figures in our manuscript to be more readable, especially Figure 1.
>
> Q1: We thank the reviewer for suggesting this experiment. For a stochastic ablation, we use nucleus sampling (top_p=0.9) with three temperature settings: low (0.2), medium (0.7), and high (1.2). Our preliminary results show that OLMo-2 performs slightly better overall on lower temperatures, and we will update the complete table in our manuscript when we finish running the experiment on a wider range of models.
>
> Table: Ablation results.
>
> | Temperature | Model      | -    | CF   | NS   | EA   | DS   | HE   | HH   |
> |------------|------------|------|------|------|------|------|------|------|
> | Low        | OLMo-2-1B  | 27.9 | 26.3 | 27.1 | 24.0 | 33.1 | 15.8 | 29.7 |
> | Low        | OLMo-2-7B  | 39.8 | 42.0 | 38.0 | 57.4 | 34.8 | 25.3 | 41.9 |
> | Low        | OLMo-2-13B | 44.0 | 45.6 | 40.5 | 46.9 | 46.8 | 35.8 | 45.9 |
> | Medium     | OLMo-2-1B  | 27.5 | 26.5 | 27.3 | 21.8 | 33.0 | 15.3 | 31.1 |
> | Medium     | OLMo-2-7B  | 38.4 | 40.6 | 38.7 | 55.6 | 29.8 | 24.5 | 41.9 |
> | Medium     | OLMo-2-13B | 43.9 | 46.7 | 41.5 | 46.4 | 47.0 | 37.6 | 45.7 |
> | High       | OLMo-2-1B  | 31.2 | 23.4 | 28.7 | 22.4 | 29.0 | 13.8 | 27.6 |
> | High       | OLMo-2-7B  | 36.6 | 36.1 | 37.2 | 48.6 | 32.8 | 23.1 | 37.0 |
> | High       | OLMo-2-13B | 40.9 | 44.5 | 39.5 | 43.8 | 44.5 | 35.0 | 43.1 |

---

> > ### Comment · Reviewer_kuNP · 2025-11-26
> > **Interesting new results**
> >
> > Dear authors,
> >
> > Thanks for your prompt response and additional experiments. It is interesting that with a bit of stochasticity, the performance of the OLMO models seem to improve? I think this partially aligns with the common intuition that higher temperature can lead to more potential hallucinations.
> >
> > Sorry I'm not super familiar with the manuscript system at ICLR -- Do we get to see the updated manuscript during rebuttal? Revisiting this paper, I think not only Figure 1, all figures need larger fonts (including axis and tick labels and such). An example paper for some ideal font-to-image ratio: https://arxiv.org/pdf/2310.01732

---

> ### Author Response · Authors · 2025-12-03
> **Thank you for your quick response**
>
> We are heartened that the reviewer finds our new results interesting. We finished running the ablation on more models and updated the full results in Table 3 of the new PDF. We find that stochasticity only has a very minor impact on critical confabulation performance (most models show a small drop on higher temperatures), and our results are largely robust to sampling temperature.
>
> We made sure to fix the figures problem is solved in our updated PDF -- all figures now have larger fonts. We thank the reviewer for giving us a helpful example!
>
> We thank the reviewer again for their helpful comment.

---

### Official Review · Reviewer_Qjc3 · 2025-11-01

**Soundness:** 3
**Presentation:** 2
**Contribution:** 2
**Rating:** 4
**Confidence:** 3

**Summary:**

This paper uses LLMs’ predictions in a narrative cloze task to show that LLMs can plausibly speculate missing information about historical figures. The authors relate this use of LLMs to “critical fabulation”, a practice in African American studies where scholars repair missing information in archives through storytelling.

**Strengths:**

I feel like this paper is so unlike most of the ICLR paper I’ve seen in recent years, especially with its use of historical archival data and its engagement with African American Studies. I also appreciate the authors’ thoughtfulness around experimental decisions.

**Weaknesses:**

Though this work is a rare interdisciplinary blend of technical and substantive work, I worry that the way the authors explain their methodology and results may be a barrier for this work actually being useful for “social good”. That is, true “social good” should make research accessible and communicable to the communities associated with the datasets involved. Otherwise, such work is at risk of being extractive. Even as someone with a technical background, I found that there were some parts of the paper that I had to read very very carefully to understand. The authors do not have to necessarily decrease the technicality of their work, but to reconsider how they communicate and illustrate their methods and key takeaways.

The authors use a lot of paper space to justify why OLMo-2 is their model that they want to focus their analysis on, and why auditing for data contamination is important, only to find out that memorization might not be an issue because OLMo-2’s performance isn’t that different from its unaudited peers. I wonder if the structure/narrative of this paper could be reorganized so that the amount of attention dedicated to certain concepts is more correlated with the potential impact of findings. The main paper is very dense already; some things could be deprioritized into the appendix.

In addition, I found that the beginning of this paper to introduce the interesting idea of “critical confabulation” and its implications, but the paper ends abruptly without reconsidering how some of these earlier themes may relate to the model-evaluation-based findings of this paper. On a related note, I am curious what differentiates a “critical confabulation” from simply a “confabulation” – how do you know that what models are predicting is in any way “critical”?

This paper, with its “social good” framing, repeatedly suggests that their use case of LLMs is desirable for humanities scholars. I totally understand why critical fabulation is useful for scholars. However, I remain unconvinced as to why humanities scholars would actually want this process to be LLM-ified, rather than relying on their own expertise and lived experiences. Is it because an LLM could be useful for proposing a range of plausible ideas? I worry that you are making a tool just to see if you can, rather than actually fulfilling a sincere need.

A minor note: the title of this paper is too vague given the paper’s scope, and may not help this paper reach relevant audiences.

**Questions:**

Could you provide some concrete examples of knowns unknowns and unknown unknowns?

How are models’ confabulations considered “evidence-bound” (line 90, line 105, line 163) if we don’t know what kind of evidence they are using for their predictions? Are models really evidence-bound if they are making up information that may not exist in their pretraining data?

The text in Figure 1 is too small to see unless one zooms in a lot in a digital pdf viewer. Ditto for axes labels in Figure 2. In Figure 1, it’s unclear what the text saying “[MASK]” is doing in the figure.

The width of Table 2 exceeds the margins of the page. Also the caption of this table could benefit from a one-sentence takeaway for people who skim the paper.

Line 97: the citation for Chambers & Jurafsky is in the incorrect format, e.g. \citep{} instead of \citet{} maybe?

Line 253: How did you know if a document was in English or not?

Line 350: What made the three annotators qualified to do this annotation task? Who were these annotators? Ditto for the annotator in line 416.

Do you have to use “hallucinations” repeatedly to describe LLM behavior in your use case? Though this word is now very common and normalized in AI research, it still anthropormophizes LMs and diminishes the words’ origins as a serious human mental illness symptom. It is jarring to see this word upheld so prominently in a “social good”-related paper.

---

> ### Author Response · Authors · 2025-11-24
> **Response to “Weaknesses”**
>
> We thank the reviewer for appreciating the novelty of our work and data!
>
> W1 (“Though this…”): We appreciate the reviewer’s careful review and engagement with our work. We agree that the presentation of our data contamination audit is overburdened by technical details and that this detracts from and obscures the main contribution of the work. We have significantly simplified the description of this audit (Sections 3 and 4), and moved a page of detail to the “Detailed Results” section of the appendix. Moreover, we have dedicated the freed-up space to better clarify what our findings mean for the humanities and to better situate and frame our contributions in existing research questions of African American studies. We thank the reviewer for this helpful feedback. It has contributed significantly to improving the clarity and readability of the manuscript and bringing forward the value of our results.
>
> We would like to highlight that the description of our careful procedures to audit for data contamination is vital to the soundness of our work. The widely reported unreliability of pretraining data detection methods led us to focus on the audited OLMo-2. After running the main experiment with multiple unaudited models as baselines, we subsequently realized that there is not, in fact, a stark performance gap. This absence of clear evidence for memorization, while a strong signal, is not evidence for its absence. So, we feel that rigour demands that we center the data contamination audit part of the paper given memorization’s potential harm to the soundness of our work.
>
> We thank the reviewer for this helpful feedback.
>
> W2 (“In addition”): “Confabulations” (Sui et al., 2024) differ from other hallucinations in that they directly mirror human-like behavior of using storytelling as a cognitive resource for sensemaking. Given this, the governance of LLM confabulation needs to take inspiration from the critical frameworks and narrative ethics that govern human storytelling (Hartman, 2008). Our proposed concept of “critical confabulation” calls for cultural guardrails for LLM confabulation that go beyond current mechanistic mitigations of hallucination and should be a key focus for future work.
>
> W3 (“This paper…”): We thank the reviewer for raising this question, and regret not addressing this in the manuscript more concretely. We have updated the manuscript to bolster this motivation.
>
> In short, we argue that the narrative understanding capabilities of LLMs, as demonstrated in our work, could make them viable tools for (1) probing large archives to surface unknown unknowns, and (2) rapidly prototyping candidate reconstructions to survey the bounds of the known unknowns. (1) is more challenging than (2), and as the first work in this area, our paper validates LLMs’ ability to do (2). We agree with the reviewer that LLMs could be useful for proposing a wide range of possibilities, but that scholars will need to use their expertise to converge on both plausible and relevant candidates or areas of interest.

---

> ### Author Response · Authors · 2025-11-24
> **Response to “Weaknesses” (continued)**
>
> W3 (continued): Take an example from our evaluation results (see figure 1). Here’s a timeline of real historical events focused on one Tyrone Guyton, who was shot and killed by police in his West Oakland home.
> | Date       | Event                                                                                                                                                   |
> |------------|---------------------------------------------------------------------------------------------------------------------------------------------------------|
> | 1973-11-01 | Three White Emeryville policemen shot Tyrone Guyton in the back, killing the 14-year-old near his West Oakland home.                                   |
> | 1974       | The three policemen who murdered Tyrone Guyton continued to escape indictment, prompting community outrage.                                             |
> | 1975       | Tyrone Guyton’s mother, Mattie Shepherd, held an Oakland press conference demanding prosecution of his police killers.                                  |
> | …          | …                                                                                                                                                       |
> | 1979-05-03 | *(last known event)* The Black Panther Party launched a recall campaign against D.A. Lowell Jensen citing his refusal to prosecute Tyrone Guyton’s killers. |
>
> The “...” indicates a set of middle events missing from the historical archive. How do we know how Mattie Shephard’s initial activism for her son developed into a full-scale campaign of the Black Panthers? LLMs, we suggest, could help scholars rapidly prototype some working hypotheses like the following:
>
> - GPT-5: “Community members gathered at Tyrone Guyton’s death site, holding a silent vigil and documenting police surveillance of mourners.” … “Community members packed an Alameda County courtroom to witness yet another grand jury decline to indict the officers who killed Tyrone Guyton.” … “Mattie Shepherd marched alongside Black Panther Party members and community allies, uniting their struggles for justice at the Jobs and Justice rally."
> - Gemma-3-27b: “A local television news crew documented a memorial service for Tyrone Guyton attended by hundreds of community members and activists.” … “Activists delivered a petition with over 10,000 signatures to Governor Brown during a Fourth of July picnic in Sacramento” … "Mattie Shepherd forged a powerful alliance with the Southern Christian Leadership Conference (SCLC), gaining national attention and bolstering the movement for Tyrone Guyton’s justice."
> - OLMo-2-32B: “Hundreds of people attended a jam-packed Oakland City Council session in support of family demands for an inquiry into the police murder of Jose Benavidez.” … “Mattie Shepherd's home was set on fire, escalating the community's concern over the treatment of Tyrone Guyton's family.”
>
> Connections between Mattie Shephard’s fight for justice for her son and the broader contexts from the model’s known history (adjacent events like the police murder of Jose Benavidez in Oakland and the Jobs and Justice rally, or organizations/figures such as the SCLC and Gov. Jerry Brown) are plausible, and would have gone unnoticed under the archive’s voluminous details if not for the use of LLMs. Our subsequent work aims to better support this type of use case by retrieving adjacent archives relevant to a fragment and using them to augment critical confabulation.
>
> Broader impact: we aim to address the gap between scholarly perspectives in black studies (e.g., major scholars in this field such as Saidiya Hartman, Fred Moten, Christina Sharpe and others.) and public-facing work (e.g., The New York Times historical and journalistic study of US slavery, the 1619 Project). LLMs could help scholars and journalists scale up their efforts to recover lost cultural histories beyond their own expertise and lived experience, and amplify their social impact to a wider audience. We’ve updated this into a new section in our Appendix, the “Humanistic Mission Statement”.
>
> The reviewer is correct that the goal of this paper was not to actually carry out the kind of archival recovery that motivates the effort. Rather, the goal was to conduct the necessary and relevant work in this area as a means to a) open up a space for viable research questions to be asked and answered by the method under evaluation, and b) carry out the necessary preliminary evaluation to secure the plausibility of the approach ex ante.
>
> We are heartened and grateful that the reviewer finds the direction proposed and evaluated in this study provocative and promising. Our aim is to follow it up with in-depth case studies that connect our preliminary work with real-world applications.

---

> ### Author Response · Authors · 2025-11-24
> **Response to “Questions”**
>
> Q1: By ‘known unknown’, we mean things that we know that we do not know (e.g., we know that we do not know the age of the universe). By ‘unknown unknown’, we mean things that we do not know that we do not know (e.g., in the 17th century, we did not know that we did not know the mass of an electron). With respect to our project, there are many examples of ‘known unknowns’ and ‘unknown unknowns’ from scholarship in African American studies:
> Known unknowns could be:
> - “Venus,” the enslaved girl who appears in the archive only as a victim in a legal proceeding, where we know that she lived and was killed but lacks any record of her voice or interior life (Hartman 2008).
> - “Molly,” an enslaved woman in colonial Barbados whose execution is carefully documented while her familial ties, motives, and perspective remain absent from the colonial record (Fuentes 2016).
>
> Whereas, ‘unknown unknowns’ would be: erased events that the public wouldn’t even know of their existence if not for the reconstruction of historical scholarship, such as the long-suppressed documentation of the 1921 Tulsa massacre and the belated recognition of NASA’s “human computers” popularized in Hidden Figures (2016). In this domain, scholars have been particularly interested in what has been called “subaltern” intellectual history – enslaved and colonized people’s own analyses and understandings of their own lived experiences (Morgan 2015). These ephemeral historical traces are rarely recorded, as the enslaved were systematically excluded from access to writing and literacy. But it would be intellectually dishonest to overlook how they interpreted and responded to the structures of slavery in their own lives. For example, enslaved mothers understood and resisted the partus law (the 1662 Virginia law making children inherit the mother’s enslaved status) in ways not explicitly recorded, yet partially recoverable in descriptions of their actions and their fears of the enslavers (Morgan 2021).
>
> In addition, there is also the category of “unknown knowns”: knowledge inferrable from existing records that is concealed by perceptions of unknowability around marginalized archives. For instance, an 18th-century Barbados court case details the illicit affair of a white couple, but the records never acknowledge the black women who were ubiquitous in that urban setting. Reading “along the bias grain” could expose this invisibility by uncovering the fact that the white man in the case secretly sent his enslaved boy servant dressed as a woman across town to deliver messages – a subterfuge that implied how commonly black women moved about the city at night (Fuentes 2016). The authorized accounts of history reject black women’s quotidian urban presence, yet the disguise (an enslaved boy passing as a black woman) is a telling indirect sign. Critical fabulation thus uncovers an unknown known: a whole sphere of black women’s urban mobility that did not make it into the archive but is largely inferrable if we make full use of what’s there. The extraction of historical knowledge from a previously unrecognized lacuna also transforms unknown unknowns (factors no prior historian thought to consider) into a series of known unknowns that scholarship can then address.
>
> We have updated the manuscript to add clarity on this point.
>
> Q2: “Evidence-bound” confabulation is differentiated from underspecified confabulation where the LLM has to make assumptions about where the unknown unknowns are themselves. In our evaluation setting, we clearly define the bounds of what the LLM is allowed to confabulate with existing evidence.
>
> Q3: We regret not refining the visual elements of Figure 1. We have updated Figure 1 to highlight the entire masked area instead of putting a [MASK] on the side.
>
> Q4: We have corrected this formatting issue.
>
> Q5: We have corrected this citation.
>
> Q6: The language of a document is a field from its metadata, we have updated the manuscript to clarify this. We thank the reviewer for bringing this to our attention.
>
> Q7: This is an important question and we thank the reviewer for noticing this oversight. The three annotators in line 350 are PhDs or professors in the humanities with extensive experience in African American studies; the annotator in line 416 is an undergraduate history major. We have updated the manuscript to make this explicit to the reader.
>
> Q8: We agree that the term “hallucination” is harmful, and appreciate that the reviewer calls attention to the important issue of anthropormophization. We used it only to contextualize our study in existing literature. We have updated the manuscript to more carefully and deliberately define “hallucinations” as “confabulations” following Sui et al. (2024).

---

### Author Response · Authors · 2025-11-24
**Timeline for uploading the updated PDF**

We really appreciate the reviewers for their time, effort, and helpful comments! We will upload an updated PDF of our manuscript that incorporates all suggested changes in a week (11/30), including the full experiment results of the ablation recommended by reviewer kuNP.

Thank you again for your help with improving the manuscript!

---

### Author Response · Authors · 2025-12-03
**New PDF Uploaded**

We just uploaded an updated version of the paper, reflecting all the changes that the reviewers requested.

We'd like to thank the reviewers again for their very useful feedback, which have helped us significantly improve the quality and presentation of our manuscript. We understand that this is becoming rare given the rapidly growing submission and reviewing loads, and we are lucky that our reviewers went the extra mile to offer us helpful and especially substantive suggestions.

Thank you for making peer review a better place!

---

### Meta-Review · Area_Chair_pRLd · 2025-12-06

**Summary:**

The paper introduces "critical confabulation," a methodology leveraging LLMs to speculatively fill gaps in historical archives, specifically within the context of African American studies. The authors operationalize this via a narrative cloze task, utilizing audited models (OLMo-2) to rigorously control for data contamination. The reviewers unanimously appreciated the novelty of the interdisciplinary approach, the social significance of the problem space, and the technical rigor regarding data contamination audits. The initial concerns primarily revolved around the presentation (density of technical details vs. humanities framing, readability of figures), the specific definitions of terms like "hallucination" vs. "confabulation," and the need for deeper qualitative analysis of failure modes.

**Reviewer Concerns:**

1. Presentation and Narrative Structure (Reviewer Qjc3): The reviewers felt the paper was overly dense regarding the data audit, obscuring the "social good" contribution. The authors have reorganized the manuscript, moving technical audit details to the appendix and expanding the discussion on the humanities implications and the "humanistic mission''.

2. Visual Legibility (Reviewers Qjc3, kuNP): Multiple reviewers noted that figure fonts were illegible. The authors have updated the figures to increase font sizes and readability, a change explicitly acknowledged and appreciated by Reviewer kuNP during the discussion.

3. Qualitative Analysis and Robustness (Reviewers kuNP, ubaQ): In response to requests for better analysis, the authors added an error taxonomy and distribution analysis. They also conducted a temperature ablation study (requested by kuNP) which demonstrated model robustness.

4. Methodological Clarifications (Reviewer ubaQ): The authors clarified the specifics of the cloze tasks and justified the empirical thresholds used for name filtering.

**Reviewer Scores:**

Reviewer Qjc3: Current Score: 4. Likely Maintain Score: 4.
Reviewer kuNP: Current Score: 6. Likely Maintain Score: 6.
Reviewer ubaQ: Current Score: 8. Likely Maintain Score: 8.

---

### Decision · Program_Chairs · 2026-01-26

Accept (Poster)